# Inorganic perovskite-based active multi-functional integrated photonic devices

Qi Han[1,6], Jun Wang[2,6] ✉, Shuangshuang Tian[2], Shen Hu[1,3] ✉, Xuefeng Wu ©[1,4], Rongxu Bai[1], Haibin Zhao ©[2], David W. Zhang[1,3,4,5], Qingqing Sun[1,3,4] ✉ & Li Ji ©[1,3,4,5] ✉

The development of highly efficient active integrated photonic circuits is crucial for advancing information and computing science. Lead halide perovskite semiconductors, with their exceptional optoelectronic properties, offer a promising platform for such devices. In this study, active micro multi-functional photonic devices were fabricated on monocrystalline CsPbBr₃ perovskite thin films using a top-down etching technique with focused ion beams. The etched microwire exhibited a high-quality micro laser that could serve as a light source for integrated devices, facilitating angle-dependent effective propagation between coupled perovskite-microwire waveguides. Employing this strategy, multiple perovskite-based active integrated photonic devices were realized for the first time. These devices included a micro beam splitter that coherently separated lasing signals, an X-coupler performing transfer matrix functions with two distinguishable light sources, and a Mach-Zehnder interferometer manipulating the splitting and coalescence of coherent light beams. These results provide a proof-of-concept for active integrated functionalized photonic devices based on perovskite semiconductors, representing a promising avenue for practical applications in integrated optical chips.

In the post-Moore era, micro-nano-integrated photonic devices have emerged as essential core components in the process of information operation and transmission[1-3]. As the demand for integration increases, traditional electronic devices have encountered several bottlenecks[4,5], such as response speed, energy consumption, modulation, and bandwidth. Due to the benefits of ultrafast propagation speed, low loss, and diverse modulation methods, photon-based circuits have the potential to overcome these limitations and enhance the functionality of electronic devices[6-9]. Macroscopic free photonic circuits (optical path) have demonstrated numerous capabilities for transmission and logic operations[10], including waveguide propagation, directional beam splitting, intensity modulation, and phase

modulation. However, fabricating highly-integrated large-scale photonic circuits remains a challenge. Consequently, on-chip integrated photonic devices and circuits have emerged as ideal solutions, garnering significant attention. Although considerable progress has been made in the research of silicon-based integrated photonic devices[11,12], efficient and low-cost active heterogeneous integration of IV-based and III-V-based light sources remains a challenge. Hence, active integrated photonic circuits utilizing novel gain materials are crucial for future integrated photonics research[13].

Over the past decade, an emerging class of direct bandgap semiconductor materials, known as lead halide perovskites, has gained widespread attention in the fields of light-emitting diodes[14-16], solar

[1]State Key Laboratory of ASIC & System, School of Microelectronics, Fudan University, Shanghai 200433, China. [2]Department of Optical Science and Engineering, School of Information Science and Technology, Key Laboratory of Micro & Nano Photonic Structures, Shanghai Frontiers Science Research Base of Intelligent Optoelectronics and Perception, and Shanghai Ultra-precision Optical Manufacturing Engineering Research Center, Fudan University, Shanghai 200433, China. [3]Jiashan Fudan Institute, Jiaxing 314110, China. [4]Zhangjiang Fudan International Innovation Center, Shanghai 201210, China. [5]Hubei Yangtze Memory Laboratories, Wuhan 430205, China. [6]These authors contributed equally: Qi Han, Jun Wang. ✉e-mail: wangjunfd@fudan.edu.cn; hushen@fudan.edu.cn; qqsun@fudan.edu.cn; lji@fudan.edu.cn

cells[17–19], and lasers[20–24], owing to their exceptional optoelectronic properties[25,26]. These properties include high photoluminescence (PL) quantum yield, low defect state concentration, long charge-diffusion length, and broad wavelength tunability. Moreover, monocrystalline perovskite microstructures featuring regular and sharp morphology have demonstrated remarkable laser and waveguide performance[27–32]. Given these advantages, perovskite semiconductors represent a promising platform for realizing micro-nano light sources and photonic devices at room temperature. Numerous strategies have been proposed for the micro-nano patterned fabrication process of single-crystal perovskite; however, research on active multifunctional integrated photonic devices and circuits remains limited[33–38]. Therefore, by combining the outstanding optoelectronic properties of perovskites with state-of-the-art micro-nano surface patterning technology, perovskite-based active integrated photonic devices may offer a promising path towards the further development of integrated photonic chips, integrated optical quantum devices, and quantum computing.

In this study, high-quality, large-area monocrystalline CsPbBr$_3$ perovskite thin films were grown using chemical vapor deposition (CVD). A top-down surface etching technique, focused ion beam (FIB), was employed to fabricate perovskite-based active micro multifunctional photonic devices, including micro lasers, waveguide couplers, beam splitters, X-couplers, and Mach-Zehnder interferometers (MZIs), on insulating mica substrates. For the micro laser, lasing from a microwire served as the light source for the etched photonic devices, exhibiting a high $Q$-factor of 2460 and high coherence with a threshold of 48.7 μJ cm$^{-2}$. The effective lasing waveguide-coupling and angle-dependent coupled propagation efficiency between two microwires were investigated at various coupling angles. The perovskite-based beam splitter demonstrated the coherent and directional separation of signals from light sources with equal allocation. Furthermore, the etched X-coupler, integrated with two distinguishable microwire light sources, showcased the independent propagation of different signals without mutual crosstalk, supporting the function of the transfer matrix. Simultaneously, the perovskite-based MZI enabled the manipulation of splitting and coalescing for coherent light beams.

The findings of this study highlight the potential of employing all-inorganic perovskite semiconductors as both light sources and operational components in integrated photonic devices. Additionally, this research puts forth a conceptual proposal for active integrated optical chips.

## Results

### Perovskite-based active multifunctional photonic devices

The primary objective of an integrated photonic circuit is to facilitate the functionality of macroscopic optical paths and the manipulation of light on a microscopic scale. Essential functions of macroscopic optical circuits, such as signal generation, directional propagation, directional beam splitting, transfer matrix, and phase modulation, can be achieved through corresponding integrated photonic devices. Inspired by this idea, perovskite-based active integrated photonic devices were designed and fabrication, including micro lasers, waveguide couplers, micro beam splitters and integrated interferometers, to demonstrate basic functionalities of optical matrix operation (allocation, transfer and modulation), as shown in schematic formulas of Fig. 1a–c and details in Supplementary Materials.

A well-established FIB etching technology enables the control of semiconductor microstructures' morphology with nanometer-scale spatial resolution and arbitrary patterning[30,31,39]. By accurately adjusting the operating voltage and current parameters, micro-nano structures with regular and smooth edges can be programmatically patterned onto the target object. This FIB technology presents a promising approach for fabricating high-quality, integrated,

multifunctional photonic devices based on perovskite semiconductors without resorting to a complex process. Scanning electron microscopy (SEM) images of the perovskite beam splitter, X-coupler, and MZI with active micro laser sources following FIB treatment are displayed in Fig. 1d, e. These microstructures are based on a large-area and high-quality monocrystalline CsPbBr$_3$ perovskite thin film. To ensure greater accuracy and minimal damage[30,31], a beam current of 40 pA is employed to etch the gaps between coupled structures (Methods), effectively separating the source from the functional part. The left etched microwire of each device can form Fabry-Pérot (FP) microcavities, which comprise opposite end facets that confine photons and facilitate lasing emission[27–29]. Consequently, these microcavities provide the source for integrated photonic devices. Furthermore, the intensity and shape of photoluminescence (PL) emission for the perovskite thin film remain nearly consistent before and after FIB treatment (Supplementary Fig. S1), thus substantiating the feasibility of directly fabricating photonic devices.

### Characterizations of monocrystalline CsPbBr$_3$ thin films

The monocrystalline CsPbBr$_3$ thin film, which serves as the raw material of photonic devices, is grown by a CVD process (Methods)[40]. The upper panel of Fig. 1g shows the fluorescence microscope image of CsPbBr$_3$ film with a thickness of 410 nm (Supplementary Fig. S2), indicating uniform fluorescence and large areas to meet the requirements of device fabrication. The brighter lines of the thin film represent the crystal boundaries formed during the epitaxial growth process[30,40]. X-ray diffraction (XRD) and transmission electron microscopy (TEM) measurements were carried out to confirm the crystal structure of CsPbBr$_3$ thin films. As shown in the lower panel of Fig. 1g, tiny splitting XRD diffraction peaks of the sample at -15.2° and 30.7° are indexed to the orthorhombic phase structure (ICSD #97851), revealing the excellent crystal quality of the thin film without any impurity peaks from CsBr or PbI$_2$ (Supplementary Fig. S2)[30,41]. Moreover, a high-resolution TEM image of the cross-section for CsPbBr$_3$ thin film demonstrates the lattice spacing of 0.29 nm, also showing the brilliant monocrystalline property (Supplementary Fig. S2). The energy-dispersive X-ray spectroscopy analysis characterizes the elemental composition of the thin film cross-section, displaying a uniform spatial distribution of Cs, Pb, and Br elements (Supplementary Fig. S2). X-ray photoelectron spectroscopy collected the signals of single Cs$_{3d}$, Pb$_{4f}$, and Br$_{3d}$ to confirm the chemical states and chemical purity of the thin film (Supplementary Fig. S3). PL and time-resolved PL (TRPL) measurements were conducted to further evaluate the optical quality of the monocrystalline CsPbBr$_3$ thin film at room temperature. The normalized PL and absorption spectra in Fig. 1h show an exciton emission peak centered at 525.2 nm with a linewidth of 14.4 nm and a strong excitonic absorption centered at 515.4 nm. TRPL delay dynamics (Fig. 1i) reveals two different lifetime components of the thin film, i.e., a fast component of 12.74 ns and a slow component of 96.52 ns. Such two different time scales could be attributed to the radiative recombination of intrinsic excitons (slow) and the bimolecular recombination process of excitons (fast) in the system[42,43]. These results exhibit high optical quality and agree with the reported literature[30,40].

### Micro laser based on perovskite microwire

In light of the exceptional optical quality of monocrystalline CsPbBr$_3$ perovskite thin films, a microwire was fabricated using FIB etching and its lasing emission quality was characterized. The etched CsPbBr$_3$ microwire exhibits a smooth surface and sharp edges with a dimension of 15 μm × 3 μm × 0.525 μm, forming a transverse Fabry-Pérot (FP) microcavity (Fig. 2a and Supplementary Fig. S4). At room temperature, under low-power excitation by a 400 nm femtosecond pulsed laser, the PL microscope image of the microwire (Fig. 2b) reveals a uniform green-color emission. Under high pump fluence and above the

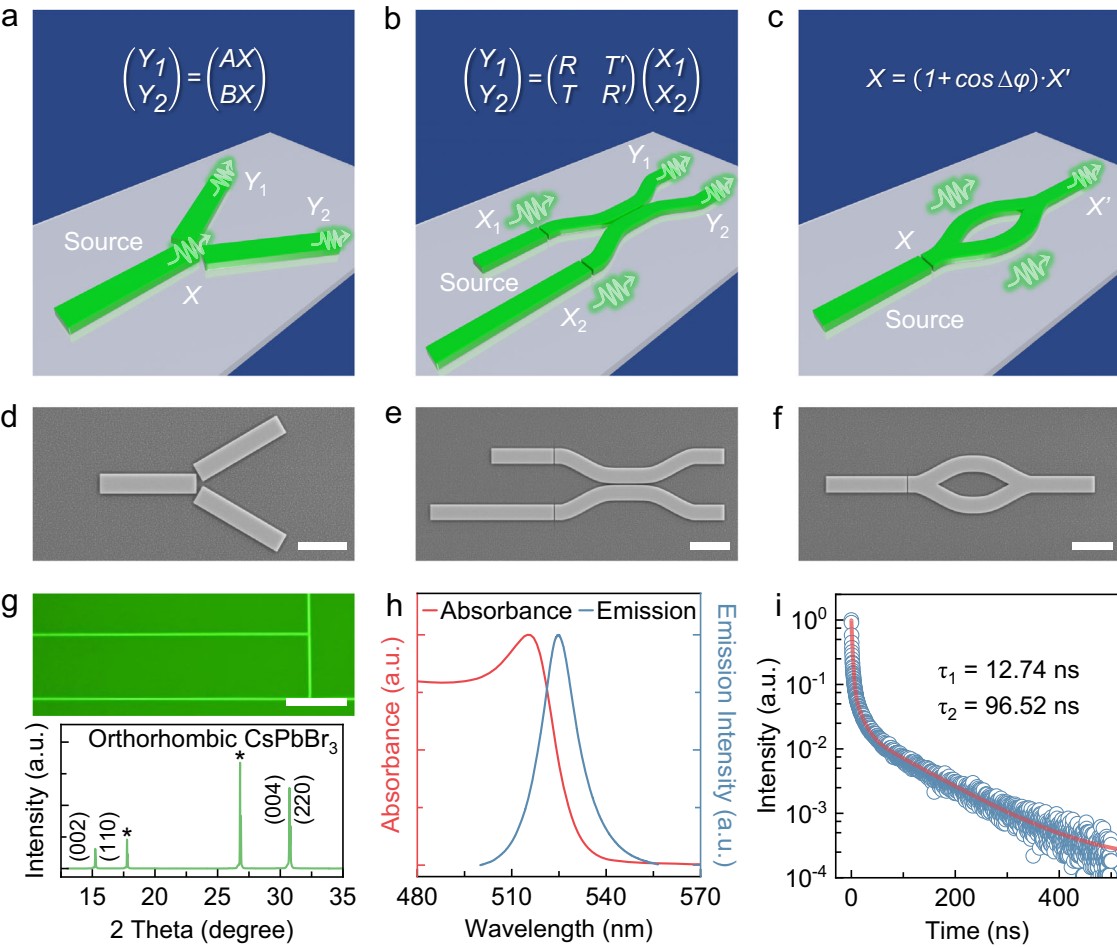

**Fig. 1 | Perovskite-based active multifunctional photonic devices and optical characterizations of monocrystalline CsPbBr₃ thin film. a–c** Schematics of perovskite active beam splitter, X-coupler, and Mach-Zehnder interferometer, respectively. **d–f** SEM images of the monocrystalline perovskite active beam splitter, X-coupler, and Mach-Zehnder interferometer after FIB treatment, respectively. Scale bar: 5 μm. **g** Fluorescence microscope image (the upper panel) and XRD (the lower panel) of a monocrystalline CsPbBr₃ thin film, showing uniform fluorescence emission and morphology and the orthorhombic phase structure. The XRD peaks originating from the pure mica substrate are marked by *. Scale bar: 30 μm. **h** Room temperature absorption (red curve) and PL emission spectra (blue curve) of a monocrystalline CsPbBr₃ thin film. **i** TRPL spectrum of the monocrystalline CsPbBr₃ thin film, the fitting (red curve) gives two lifetime components of 12.74 and 96.52 ns.

threshold, strong lasing emissions are observed to distinctly leak out from the opposite end facets of the microwire (Fig. 2c), attributable to the FP-mode oscillation[27–29].

Numerical simulations further substantiated the lasing formation of the FP oscillation ($\lambda = 538.63$ nm, $n = 2.53$) and displayed a typical transverse FP standing-wave mode in the normalized electric field intensity distribution along the microwire (Fig. 2d), in agreement with experimental results[23,28]. The PL emission evolution was examined as the pump fluence increased (Fig. 2e). At a low pump fluence of 46 μJ cm⁻², a broad spontaneous emission (SE) with a linewidth of 15.9 nm is observed. With the pump fluence increasing further above 48.7 μJ cm⁻², several sharp peaks of lasing emission abruptly emerge at the low-energy side of SE, becoming dominant in the PL spectra as the intensity rapidly rises. The inset in Fig. 2e illustrates the Lorentz fitting of one magnified lasing oscillation mode with a linewidth of 0.219 nm and a $Q$-factor of 2460 at 55 μJ cm⁻². The evolution of the integrated PL intensity and linewidth as functions of pump fluence reveal a nonlinear process of the lasing emission in the microwire (Fig. 2f). When the pump fluence surpasses the critical threshold (48.7 μJ cm⁻²), the linewidth dramatically decreases from 15.9 nm to 0.219 nm, and the intensity exhibits a typical S-shaped curve with an increase of nearly four orders of magnitude, indicating the transition from SE to lasing emission[27–29].

Moreover, angle-resolved PL (ARPL) measurements were conducted to investigate the $k$-space (far-field) information of the lasing emission. Above the threshold, the ARPL spectrum of the CsPbBr₃ microwire unambiguously demonstrates that several standing-wave-like interference patterns extend to all detection angles on lasing modes (Fig. 2g). These patterns originate from the interference between coherent lasing modes emitted from the opposite end facets of the microwire[23]. To further investigate the cavity geometry-dependent lasing performance, we measured the lasing properties of the multiple different length and width microwires as well as an etched microdisk (Supplementary Figs. S5–S7). These results demonstrate that the lasing performance of the etched CsPbBr₃ laser can be precisely controlled by tuning the geometry of the microcavity. Thus, a high-quality, coherent and stable microwire laser can be achieved through in-situ FIB etching on demand (Supplementary Fig. S8).

**Lasing propagation and waveguide coupling in microwires**

In order to investigate the propagation characteristics of perovskite microwire lasing for integrated multifunctional photonic devices, waveguide-coupling measurements were conducted at various coupling angles between two etched CsPbBr₃ microwires. In these configurations, one microwire serves as the lasing source, while the other functions as the propagation medium, with coupling angles of 0°, 30°,

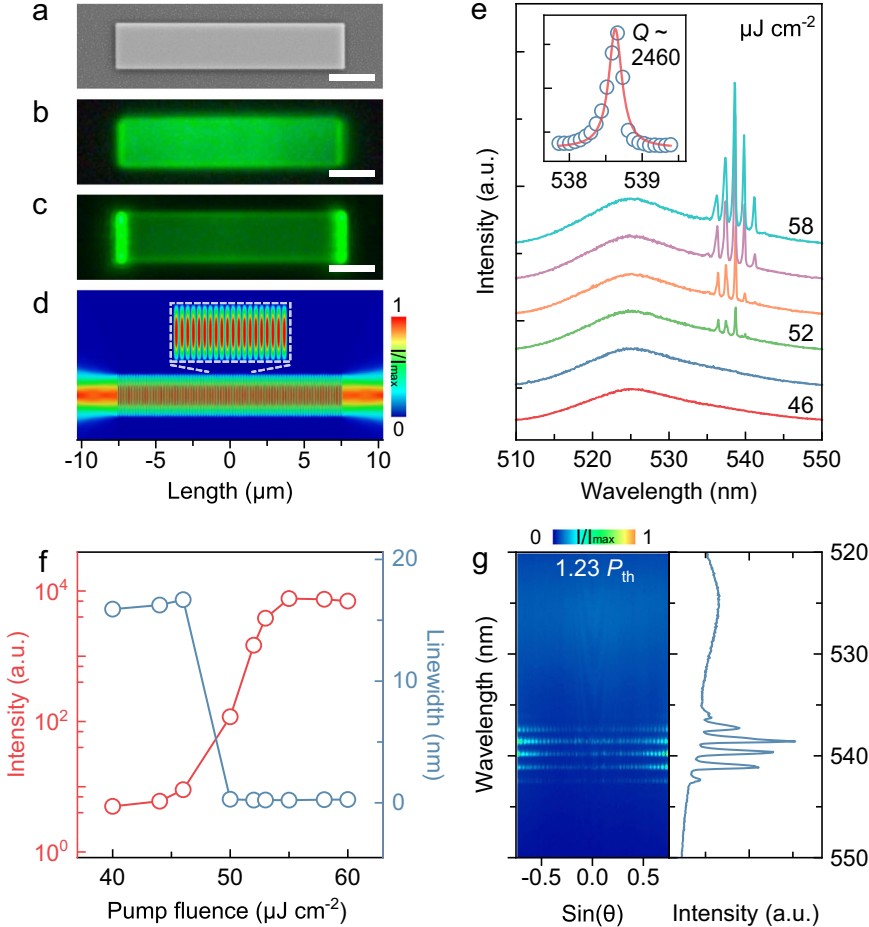

**Fig. 2 | Lasing characterizations of a monocrystalline CsPbBr₃ microwire. a** SEM image. Scale bar: 3 μm. PL microscope images below (**b**) the threshold (0.92 $P_{th}$) and above (**c**) the threshold (1.23 $P_{th}$), respectively. Scale bar: 3 μm. **d** FDTD simulation of the 2D normalized electric field intensity distribution along the microwire ($\lambda$ = 538.63 nm, $n$ = 2.53), defining a FP cavity standing-wave mode. **e** The PL spectra emitted from the microwire with the pump fluence increasing from 46 to 58 μJ cm⁻². The inset shows one magnified lasing oscillation mode with a linewidth of 0.219 nm and a $Q$-factor of 2460 above the threshold (at 55 μJ cm⁻²). **f** The evolution of the integrated PL emission intensity (red curve) and linewidth (blue curve) as functions of pump fluence of the microwire, showing the threshold of 48.7 μJ cm⁻². **g** ARPL (left) and PL (right) spectra of the microwire above the threshold (1.23 $P_{th}$). The patterns of the ARPL spectra originate from the interference of coherent lasing modes emitted from two edges of the such microwire.

60°, and 90°, respectively. Such configurations have been shown to provide explicit demonstrations of lasing propagation and waveguide coupling[44,45].

Two microwires of identical dimensions (10 μm × 2 μm × 0.515 μm) are separated by a 250 nm gap (refer to the left insets of Fig. 3a–d and Supplementary Fig. S9), exhibiting a uniform and robust fluorescence emission (Supplementary Fig. S10). When excited above the threshold, the upper microwires emit lasing, and the propagating coupled signals are detected at the terminals of both microwires to compare the PL spectra of input and output. At a pump fluence of 65 μJ cm⁻² (above the threshold), the real-space PL images of waveguide couplers reveal that the intensity and direction of lasing propagation can be directly modulated by the coupling angle. The PL spectra detected at points A and B exhibit the same wavelength and shape for all lasing peaks at each coupling angle (Fig. 3a–d), indicating efficient light coupling from the lasing sources to point B. The coupled propagation efficiencies, defined as the ratio of lasing integral intensity for points B and A ($T_{AB}$), are calculated as 38.55%, 17.09%, 7.65%, and 7.50% for coupling angles of 0°, 30°, 60°, and 90°, respectively, exhibiting a dependence on the angle. The observed propagation loss could be primarily attributed to the extended propagation distance and self-absorption, analogous to that in III-V-based integrated photonic circuits[46,47]. Additionally, the pump fluence dependence of

propagation efficiency for a coupling angle of 0° displays an increasing trend from 20.88% to 39.00% as the pump fluence increases (Supplementary Fig. S11). The gap-dependent lasing waveguide-coupling measurements were also performed in two coupled microwires with different coupling gaps (Supplementary Fig. S12). With the gap distance increasing, the propagation efficiency for lasing signals displays a decay trend. A similar phenomenon is showcased in the multi-stage lasing waveguide-coupling measurements (Supplementary Fig. S13). These findings underscore the effective waveguide coupling and propagation of lasing emission between two coupled perovskite microwires.

**Perovskite-based active beam splitter**

Directional beam splitting is crucial for multifunctional integrated photonic devices, we thus design a perovskite beam splitter with an integrated lasing source based on effective waveguide coupling. The etched beam splitter is composed of three CsPbBr₃ microwires with the identical size of 10 μm × 2 μm × 0.460 μm, where the horizontal wire works as a lasing source and the other two symmetric arms with an angle of 60°, and a coupled gap of 233 nm with the source (Fig. 1d and Supplementary Fig. S14). Such perovskite beam splitter could support a strong lasing emission and effective waveguide coupling to realize equal allocation of light on microscale[13]. Figures 4a, b display

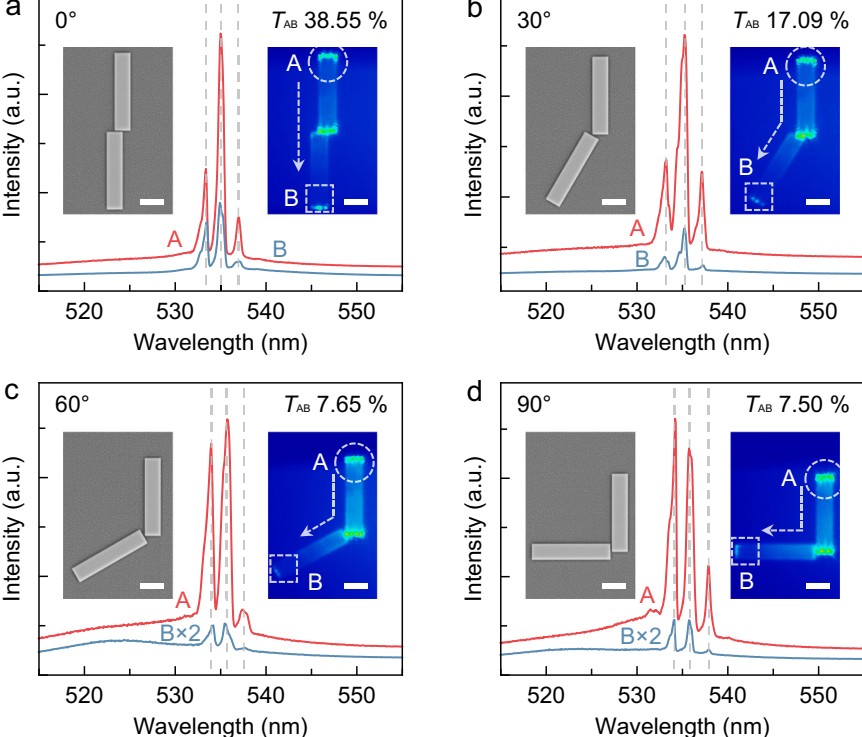

**Fig. 3 | Waveguide-coupling lasing propagation based on two monocrystalline CsPbBr₃ microwires.** PL spectra emitted from two edges of the waveguide couplers at different coupling angles of 0° (**a**), 30° (**b**), 60° (**c**), and 90° (**d**), respectively, above the threshold (at 65 μJ cm⁻²). The red curves and blue curves represent PL emissions detected from the regions of excitation terminals (A points) and propagation terminals (B points), respectively. The diameter of the excitation spots is about 10 μm. The left insets are SEM images of the waveguide couplers with a gap of 250 nm between the two microwires. The right insets are real-space PL images of such waveguide couplers under the pump fluence of 65 μJ cm⁻². Scale bar: 3 μm. The white dashed circles and boxes depict the regions of excitation terminals and propagation terminals, respectively. The dotted arrows represent the directions of lasing propagation. The propagation efficiencies ($T_{AB}$) are 38.55%, 17.09%, 7.65%, and 7.50% for the coupling angles of 0°, 30°, 60°, and 90°, respectively.

the real-space PL image and the PL microscope image of the beam splitter when the source is pumped with 63 μJ cm⁻². One can observe that the FP lasing emission is separated and propagated into two arms by directional waveguide coupling, subsequently the approximate equal-proportional lasing modes output from two terminals (C and D points). Figure 4d shows the normalized PL spectra detected at the terminal A, C and D points, demonstrating that the input and output share the same signal, due to the consistent lasing shape and wavelength (at 534.34 and 536.09 nm). To confirm the equal-proportion allocation of light, the propagation efficiencies of two arms ($T_{AC}$ and $T_{AD}$) are calculated as 15.98% and 15.62%, respectively. To prove the coherence of the output signal for the two arms, ARPL spectra (Fig. 4e) were performed to simultaneously collect the far-field emission of C and D points, as shown in the white dashed box of Fig. 4b. The standing-wave-like patterns with equal space are obtained on the k-space image of Fig. 4e, as direct evidence for the interference of coherent lasing modes emitted from two splitter terminals[23].

In addition, we utilized the other excitation configuration to realize beam splitting of lasing modes in unequal proportion (Fig. 4c). In this scenario, the lower wire as a lasing source is excited with 63 μJ cm⁻², then the lasing modes are split and propagated into A and C points, where two arms are coupled with the source at an asymmetrical angle. Normalized PL spectra detected at the terminal D, A and C points (Fig. 4f) exhibit the same peak position of lasing modes with unequal propagation efficiencies ($T_{DA}$ = 4.50% and $T_{DC}$ = 1.38%). The allocation proportion of the beam splitter can be modulated by the coupling angle between the source and waveguide arm. Thus, our perovskite-based active beam splitter demonstrates an expected performance that can realize on-demand directional beam splitting of light on the chip.

## Perovskite-based active X-coupler and MZI

The elements of photonic devices including microwire laser, waveguide coupler, and beam splitter have been successfully fabricated using the FIB process. This has enabled a proof-of-concept demonstration of active integrated photonic devices on patterned monocrystalline perovskite thin films. The splitting and coalescence of multiple light beams are essential for integrated photonic logical circuits, where X-coupler and MZI serve as prototypical structures for applications in optical modulation[1,6,48]. In this study, perovskite-based active X-coupler and MZI were employed to realize the conceptual function of the transfer matrix on the photonic chip.

As depicted in the SEM image in Fig. 1e, the etched active X-coupler consists of microwire lasers and curved waveguides and exhibits strong emission under a fluorescence microscope (Supplementary Fig. S10). Two microwire lasers of differing dimensions (upper: 8 μm × 2 μm × 0.5 μm, lower: 16 μm × 2 μm × 0.5 μm, Supplementary Fig. S15) provide lasing emissions with distinguishable mode positions, functioning as two signal sources. The lasing emission signals are in-plane coupled and guided along the curved waveguides. Owing to the cross-waveguide coupling between the two curved wires, the output signals of the X-coupler facilitate the operation of the transfer matrix. In this structure, subwavelength gaps of 122.5 nm (G1 and G2) and 210.7 nm (G3) separate the source and the two curved waveguides (Supplementary Fig. S15). The output signals (C and D points) under varying input conditions are measured to demonstrate sufficient propagation and to obtain the elements of the transfer matrix. For the single G1 input case (Fig. 5a), the upper left microwire is excited above the threshold (at 65 μJ cm⁻²) to generate lasing modes, which are coupled with the X-coupler through the G1 gap. The X-coupler divides the coupled signals into two so-called reflection and

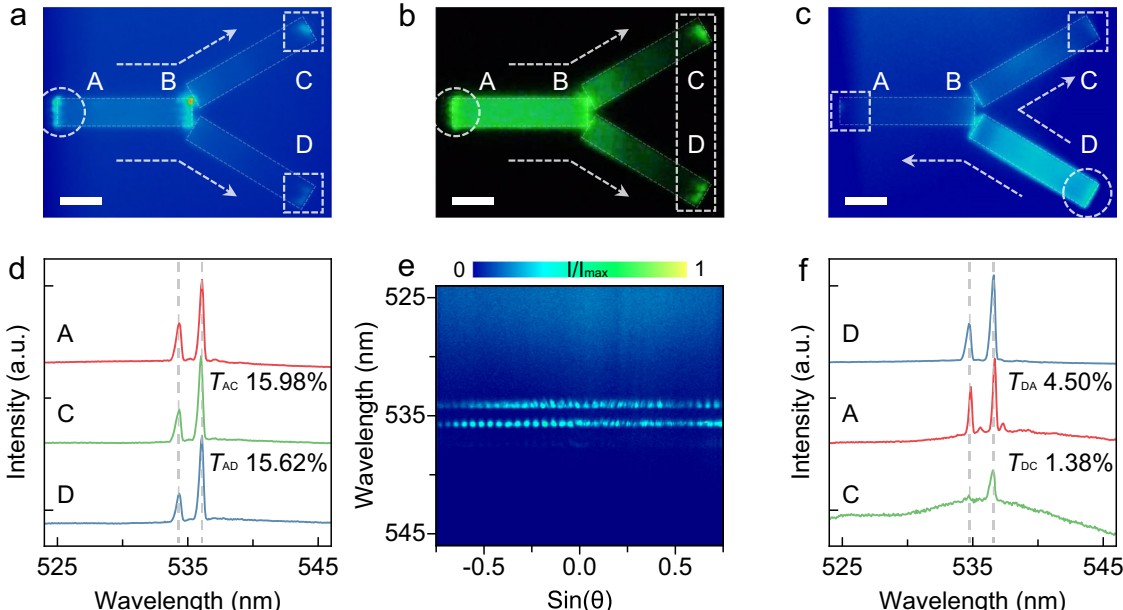

**Fig. 4 | Characterizations of the CsPbBr₃ beam splitter.** Real-space PL images (**a**, **c**) and PL microscope image (**b**) of the perovskite beam splitter above the threshold (at 63 µJ cm⁻²). The white dashed circles and boxes depict the regions of excitation terminals and propagation terminals, respectively. The diameter of the excitation spots is about 10 µm. The dotted arrows represent the directions of lasing propagation. Scale bar: 3 µm. **d, f** Normalized PL spectra of the perovskite beam splitter with different excitation configurations and propagation directions corresponding to (**a**) and (**c**). The red, green and blue curves represent the PL emission detected at A, C, and D points, respectively. The propagation efficiencies $T_{AC}$ and $T_{AD}$ for excitation configuration (d) are 15.98% and 15.62%, and $T_{DA}$ and $T_{DC}$ for excitation configuration (f) are 4.50% and 1.38%, respectively. **e** ARPL spectra of the beam splitter detected at both C and D points simultaneously, exhibiting the coherence of lasing emission from the two arms of the beam splitter.

transmission beams via the G3 gap, characterized by the reflection coefficient $R$ and the transmission coefficient $T$, respectively. In Fig. 5c, the normalized PL spectra individually detected from A, C, and D points exhibit the consistent shape and wavelength of lasing modes (at 535.33 nm and 537.58 nm), with the extracted reflection coefficient $R_{AC}$ of 12.08% and transmission coefficient $T_{AD}$ of 1.61%, indicating the clear consistency of the outputs and the source.

In the single G2 input case (Fig. 5b), a similar phenomenon of signal propagation is observed, with reversed outputs yielding $R_{BD}$ of 10.12% and $T_{BC}$ of 1.09% above the threshold (at 59 µJ cm⁻²), as illustrated in Fig. 5d. Consequently, the transfer matrix of the active X-coupler is obtained:

$$\begin{pmatrix} 12.08\ \% & 1.61\ \% \\ 1.09\ \% & 10.12\ \% \end{pmatrix}.$$

In the dual-source input configuration (Fig. 5e), two sets of distinguishable lasing signals are introduced simultaneously to the X-coupler above the threshold (at 62 µJ cm⁻²). The PL spectra, detected at points A, B, C, and D, explicitly reveal that the outputs at points C and D are mixed with two sets of different signals. These signals undergo antisymmetric reflection and transmission coefficients and originate from two microwire lasers (Fig. 5g). These findings demonstrate the independent propagation of distinct signals without mutual crosstalk in the perovskite-based X-coupler.

Furthermore, MZI integrated with a microwire lasing source is designed to demonstrate the coalescence of coherent light beams in the perovskite-based photonic device. The microwire laser (10 µm × 2 µm × 0.5 µm) and the etched MZI are coupled via a gap (G4) of 122.5 nm (Fig. 1f, Supplementary Fig. S10 and S16). Lasing signals are obtained above the threshold (at 63 µJ cm⁻²), subsequently split and coalesced along the two arms of the device, and finally output from point D (Fig. 5f)[13,48]. Normalized PL spectra are individually detected at points A, C, and D, maintaining consistent shape and wavelength (at

535.73 nm, 537.41 nm, and 539.45 nm), with the propagation efficiencies $T_{AC}$ and $T_{AD}$ extracted as 8.46% and 7.28%, respectively (upper panel of Fig. 5h). These results demonstrate the potential of the device for manipulating coherent light beams.

Meanwhile, ARPL spectra measurements were performed at the coalescence region to prove the coherence of the two lasing signals, as marked in the white dashed box of point C (Fig. 5f). Standing-wave-like patterns on the $k$-space image unambiguously indicate the coherence of the lasing signals leaked from the two arms (lower panel of Fig. 5h)[23]. The results confirm that active, integrated functionalized photonic devices can be achieved using perovskite semiconductors, further broadening the material systems of integrated optical chips.

## Discussion

In summary, the etched microwire, characterized by its smooth surface and sharp edge, exhibits a highly coherent micro laser with transverse FP modes and a high $Q$-factor of 2460, thus providing a strategy for integrated light sources. The effective lasing waveguide coupling and propagation between coupled perovskite microwires have been verified to be feasible, with coupled propagation efficiencies calculated at 38.55%, 17.09%, 7.65%, and 7.50% for coupling angles of 0°, 30°, 60°, and 90°, respectively. Additionally, the fabricated active beam splitter effectively separates lasing signals from light sources, with an equal allocation of 15.98% and 15.62% on the microscale. Furthermore, the perovskite X-coupler, integrated with two distinguishable microwire light sources, demonstrates the anticipated performance, which includes the independent propagation of distinct lasing signals without crosstalk and the functionality of the transfer matrix. The perovskite-based MZI coherently divides and combines light beams on the microscale, and numerous modulations could potentially be achieved within such a structure.

Perovskite semiconductors have been identified as promising candidates for the realization of multifunctional integrated optical devices, owing to the exceptional optical quality of monocrystalline

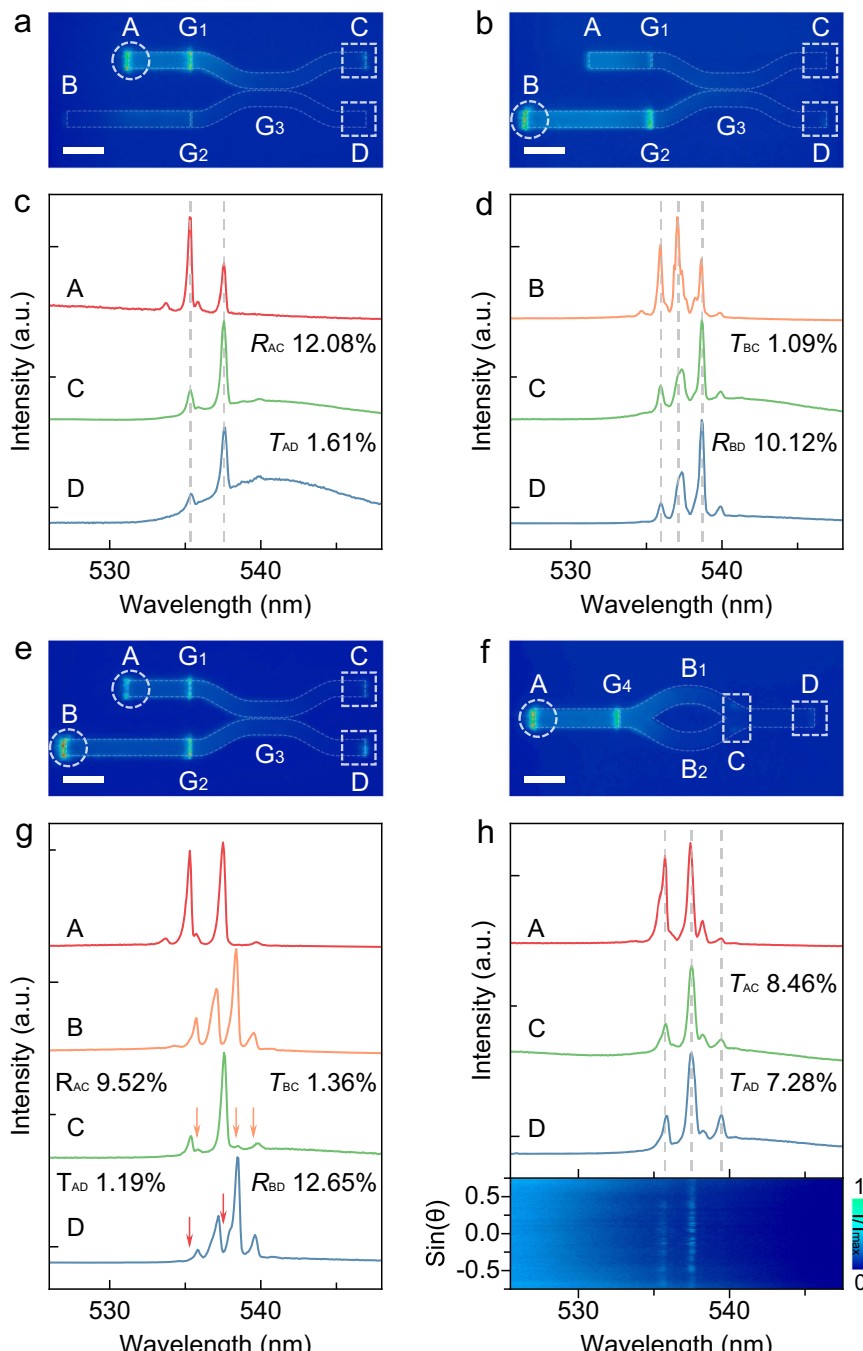

**Fig. 5 | Characterizations of the CsPbBr$_3$ X-coupler and MZI.** Real-space PL images of the CsPbBr$_3$ X-coupler and MZI above the threshold with different excitation configurations, i.e., the excitation at A point for (**a**), at B point for (**b**), at both A and B points for (**e**), and at A point for (**f**), respectively. Scale bar: 5 µm. The white dashed circles and boxes depict the regions of excitation terminals and propagation terminals, respectively. The gaps G$_1$, G$_2$, G$_3$, and G$_4$ in these structures are 122.5, 122.5, 210.7, and 122.5 nm, respectively. **c**, **d**, **g** Normalized PL spectra of the X-coupler corresponding to the excitation configurations of (**a**), (**b**), and (**e**), respectively. The red, orange, green and blue curves represent the PL emission detected at A, B, C, and D points, respectively. The red and orange solid arrows point out the cross-coupled lasing propagation modes in the outputs. **h** The upper panel: Normalized PL spectra of the MZI corresponding to (**f**). The red, green and blue curves represent the PL emission detected at A, C and D points, respectively. The propagation efficiencies ($T_{AC}$ and $T_{AD}$) are 8.46% and 7.28%, respectively. The lower panel: ARPL spectra of the MZI detected at C point, showing the coherence of the propagated lasing modes from two arms of the interferometer.

CsPbBr$_3$ thin films and the precision of FIB etching technology. A range of perovskite-based active micro photonic devices have been successfully implemented, including micro lasers, waveguide couplers, beam splitters, X-couplers, and MZIs, which together may constitute a conceptual photonic chip. These results pave a promising path towards the development of active integrated functionalized photonic devices based on perovskite semiconductors and the application of specific photonic chips through a cutting-edge integration process.

In the future, integrated photonics will develop towards subwavelength nanostructures. Bound-states-in-the-continuum and moiré photonic crystals could facilitate on-chip vortex and high-$Q$ laser sources[49,50]. Light-matter strong coupling could also generate room-temperature ultralow threshold polaritonic lasers in nanocavities[51].

Metasurface structures could manipulate highly efficient in-plane propagation and coupling of light[52]. The potential of nano and micro photonic devices is promising for advancing information and computing science.

## Methods

### Synthesis of monocrystalline CsPbBr₃ semiconductor

The monocrystalline $CsPbBr_3$ thin film was synthesized by a CVD method. The CVD system consists of a quartz tube furnace, an inert gas mass flowmeter and a vacuum pump. A boat was put in the heating center of the quartz tube furnace, loaded with mixed powder as the vapor source, with a mica substrate placed on one side of the tube. The compositions of the vapor source were CsBr (99.999%, trace metals basis, Sigma-Aldrich) and $PbBr_2$ (99.999%, trace metals basis, Sigma-Aldrich) powers with a 1:1 molar ratio. Before heating, the system was pumped down to 100 Pa and flushed with inert gas. Then the temperature of the furnace was raised to 585 °C and maintained for 15 mins with the gas flowing rate of 30 sccm. Finally, the furnace was naturally cooled down to room temperature.

### Fabrication of perovskite photonic devices

The $CsPbBr_3$ integrated photonic devices were fabricated by FIB (ThermoFisher Helios G4 CX, Dual Beam system) treatment on the monocrystalline $CsPbBr_3$ thin film on mica substrate. For high accuracy and smooth device surface, the major structures of the devices were etched by beam current of 80 pA, and the gaps between structures were etched by beam current of 40 pA, respectively, with the same ion beam voltage of 30 kV.

### Morphology characterizations

The morphologies of the fabricated monocrystalline perovskite microwire and photonic devices were performed by the SEM (Zeiss, GeminiSEM 300) at an operating voltage of 10 kV. The crystalline phase of the monocrystalline $CsPbBr_3$ thin film was acquired by a high-resolution X-ray diffractometer (PANalytical EMPYREAN) using Cu-Kα radiation ($\lambda = 1.5406$ Å) at room temperature.

### Optical characterizations

The fluorescence images of the monocrystalline perovskite thin film and the fabricated photonic devices were obtained through an Olympus microscope, where the sample was illuminated by an Olympus U-HGLGPS lamp. The absorption spectrum of the monocrystalline perovskite thin film was measured by a UV–VIS-NIR spectrophotometer (Agilent, Cary 5000) at $\lambda = 350-800$ nm at room temperature. The ARPL spectra, real-space PL mappings and steady-state PL spectra were measured by a Fourier imaging configuration at room temperature. The monocrystalline perovskite thin film, microwire and photonic devices were non-resonantly excited by a femtosecond pulsed laser (wavelength: 400 nm, repetition rate: 1 kHz, pulse width: 100 fs). The pump laser was focused by a microscopy objective (50×) to a 5 - 15 μm spot to locally excite the samples. The PL emission was collected by the same objective and gathered in a spectrometer with a 600 lines/mm grating and an array charge-coupled device. The TRPL spectrum of the monocrystalline perovskite thin film was measured by a time-correlated single photon counting system. All of the optical experiments were carried out at room temperature.

### Theoretical simulations

Finite difference time domain method simulation of the 2D normalized electric field intensity distribution along the perovskite microwire ($\lambda = 538.63$ nm, $n = 2.53$) was carried out by the software FDTD Solutions.

## Data availability

The data that support the findings of this study are available from the corresponding authors on request.

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

## Acknowledgements

The authors gratefully acknowledge the financial support from National Natural Science Foundation (62004044, 62204048, 12204111), State Key Laboratory of ASIC & System (2021MS004), National Key Research and Development Program of China (2022YFA1403602, 2021YFB2801804), Shanghai Science and Technology Committee Rising-Star Cultivation Program (22YF1402600), and Shanghai Pilot Program for Basic Research (22JC1403202).

## Author contributions

Q.H. grew perovskite samples and fabricated photonic devices. Q.H., S.T. and J.W. performed the experimental measurements. Q.H., J.W., S.T., J.L., S. H., X.W, R.B, H.Z., Q.S. and D.Z. analyzed the data. Q.H., J.W. and L.J wrote the manuscript with contributions from all authors.

## Competing interests

The authors declare no competing interests.
