## [Peer Review File · Nature Communications]

Inorganic Perovskite-Based Active Multifunctional Integrated Photonic DevicesREVIEWER COMMENTS

Reviewer #1 (Remarks to the Author):

In this work, Han and co-authors report integrated photonic devices based on monocrystalline CsPbBr₃ perovskite thin films. The perovskite devices were fabricated with focused ion beams. The authors have carried out systematical characterizations on the CsPbBr₃ and nanostructures via SEM, XRD, TEM and spectroscopies. The devices have also been fully characterized. Overall, this manuscript is well organized and the topic is interesting. However, the idea of perovskite based nano/micro photonics is not novel and it has been reported by multiple groups [Nano Lett. 2018, 18, 11, 6915–6923; Adv. Mater., 29 (2017), Article 1604268; Nat. Photonics, 14 (2019), pp. 82–88]. The perovskite, CsPbBr₃, has also been reported for numerous light emitting devices. In my view, the manuscript at its present state is of low scientific quality, and it lacks of enough novelty to be published in Nature Communications. I cannot at this point recommend acceptance and publication. I hope the authors find the specific questions below of use in reframing their arguments.

1. The PL spectrum shown in Figure 1h is not symmetric. Why? It seems that the perovskite microstructures showed significant amount of trap states near the band edge, and the PL spectrum showed a tail in the long-wavelength region.
2. The TRPL lifetime is too short compared with high-quality CsPbBr₃ single crystals. The lifetime of ~5 ns indicates a fast recombination process, which could be caused by the trap states as we observed from the PL spectrum. In addition, the authors cannot attribute the two lifetimes to two different recombination processes and cited two papers. The physical meaning is not sound. There are also many reports on CsPbBr₃ perovskites, which showed mono-exponential decays.
3. The authors should provide the PLQY value of these samples to show the quality of the obtained CsPbBr₃ microstructures. Ideally, the light-intensity dependent PLQY should be presented for the samples with and without patterning.
4. The authors discussed and designed the optical cavity. Hence, a cavity geometry dependent device performance should be provided. I mean, the authors should give a full comparison how the cavity geometry affects the light emission.
5. The FDTD simulation of the 2D normalized electric field intensity distribution along the microwire is meaningless. Any light emitting medium in the microwire should result similar pattern. It will be more helpful, if the authors could give more information on how to design a decent cavity, and discuss how other parameters affect the device performance. For instance, the charge carrier lifetime, the film surface roughness and etc.
6. The device performance should be compared with other perovskite micro-cavities and other conventional semiconductors, ideally in a table.
7. I am not sure if it is appropriate to term the devices as "laser". Some of the papers in the field also term them as "amplified spontaneous emission (ASE)". The observed small line width of these perovskite samples under high light excitation showed multiple modes, and there are also many emitted photons showed different energies. It seems that they are not proper laser devices at current stage.

Reviewer #2 (Remarks to the Author):

The manuscript entitled "Inorganic Perovskite-Based Active Multifunctional Integrated Photonic Devices", presents experimental demonstrations of active multifunctional integrated photonic devices based on all-inorganic perovskite, including microlasers, micro beam splitters, X-couplers, and Mach-Zehnder interferometers. Overall, this study is likely to be of interest to the broad readership of Nature Communication and contribute to the general research community in this field. However, several issues should be addressed by the authors prior to recommending the manuscript for the publication stage.

Additional technical comments are as follows:

(1) The authors chose mica as substrates for the deposition of perovskite thin films, the effect of the substrate on the crystal quality of the perovskite thin films should be discussed.

(2) Since the crystal quality of the perovskite thin films is very important for the properties of the devices, a more comprehensive analysis of the surface elemental composition and chemical state of the monocrystalline perovskite thin films is recommended.

(3) The high energy of the electron beam during FIB may have some effect or even damage on the perovskite films, the authors should discuss the effect of FIB on the PL lifetimes of perovskite, i.e., including PL results before and after FIB etching process.

(4) Since the stability is always a major concern for halide perovskite materials, it would be informative to investigate whether high pump excitations of microwire laser will change the lasing properties of the microlasers, such as the lasing threshold, and peak wavelength.

(5) It is nice that the authors have demonstrated a Mach-Zehnder interferometer (MZI) integrated with a microwire lasing source, the potential modulation properties of this MZI modulator prototype for future applications should be discussed. In addition, the microwire laser and the etched MZI are coupled via a gap of 122.5 nm, the effect of this gap should also be discussed.

(6) It seems that the coupling efficiency between perovskite microwires is not as high as expected. In practical waveguide application, multiple stages couplings are involved. How about the coupling efficiency of multi-stage waveguides? If the coupling efficiency is low, could the authors suggest potential approaches to improve this in the future?

Reviewer #3 (Remarks to the Author):

The manuscript by Qi Han et al reports the realization of integrated photonic devices based on CsPbBr₃ halide perovskites. Using focused ion beam, large area monocrystalline perovskite layers, grown by CVD on mica, are structured to obtain micrometer sized devices. Specifically, the authors demonstrate a beam splitter, Mach-Zehnder-type interferometer, and crossed-coupler. They also report characterization of lasing of a perovskite microwire and investigate the propagation efficiency of the emitted light across waveguide junctions under different coupling angles. Overall the results are a very interesting and timely as they offer proof-of-concepts for the use of CsPbBr₃ as active material in various integrated microphotonic devices. In my opinion, this is a valuable contribution and could be published after revisions. The following points should be addressed before a final decision:

1. The features in the angle-resolved plots are difficult to distinguish with the current choice of the colorscales. This is especially the case for the ARPL of the Mach-Zehnder interferometer in Fig 5h lower panel, where one cannot really recognize much of the interference pattern from the blue background. Please correct. Also Fig 2g showing angle-dependent lasing from the perovskite microwire could be improved.

2. To make the work more accessible to a broader audience of non-experts, I suggest to provide a concise, more general introduction of the basic function of the fabricated photonic devices.

3. On page 5 the authors write "TRPL spectrum" when instead referring to the TRPL decay dynamics. Same in the caption of Fig 1i. Can the authors comment in more detail on the origin of the "different recombination processes" giving rise to the bi-exponential PL decay?

4. Please revise the sentence at the end of page 10 ("Since perovskite semiconductors have been identified ..."). It seems something is missing.
5. There is a typo in the caption of Figure 1 on lines 3 and 4: "Mach-Zander". Same in the Supplementary Fig S5, last line of the caption.
6. The acronyms (FIB, PL, FP, MZI, SEM, ...) are defined multiple times throughout the manuscript.
7. Finally, in the Discussion section, I recommend discussing in more detail the relevance of the demonstrated perovskite photonic devices also in the framework of efforts towards nanophotonic design strategies to engineer light-matter interaction.

Response to Reviewers' Comments and Revised Details

Manuscript ID: NCOMMS-23-15734

Reply to Reviewer #1

Comments 1:

In this work, Han and co-authors report integrated photonic devices based on monocrystalline CsPbBr₃ perovskite thin films. The perovskite devices were fabricated with focused ion beams. The authors have carried out systematical characterizations on the CsPbBr₃ and nanostructures via SEM, XRD, TEM and spectroscopies. The devices have also been fully characterized. Overall, this manuscript is well organized and the topic is interesting.

Response 1:

We sincerely thank the reviewer for his/her time and efforts in examining our work, as well as the highly positive evaluations of our work. We also thank the reviewer for the valuable comments which are helpful to improve the quality of our manuscript. We have carefully accommodated his/her comments by performing lots of further experiments and highlighting the importance of our work. Thus, we believe that the quality of the revised manuscript has been enhanced, which is suitable for publication in *Nature Communications*.

Comments 2:

However, the idea of perovskite based nano/micro photonics is not novel and it has been reported by multiple groups [Nano Lett. 2018, 18, 11, 6915–6923; Adv. Mater., 29 (2017), Article 1604268; Nat. Photonics, 14 (2019), pp. 82-88]. The perovskite, CsPbBr₃, has also been reported for numerous light emitting devices. In my view, the manuscript at its present state is of low scientific quality, and it lacks of enough novelty to be published in Nature

Communications. I cannot at this point recommend acceptance and publication. I hope the authors find the specific questions below of use in reframing their arguments.

Response 2:

We thank the referee for allowing us to explain the new aspects and importance of our work. We have carefully read these papers [*Nano Lett.* 18, 6915 (2018); *Adv. Mater.* 29, 1604268 (2017); *Nat. Photonics* 14, 82 (2019)]. The following results are reported:

- (1) Reference of *Nano Lett.* 18, 6915 (2018) demonstrates that polycrystalline perovskite microdisk lasers with a threshold of $4.7 \mu\text{J cm}^{-2}$ and a Q-factor of 709 are coupled with the silicon nitride microwire. Such device structure belongs to heterogeneous integration and heterolayer structures, which require multi-step micro-fabrication processes and result in low coupling efficiency of cross-layer and cross-waveguide. Due to the influence of polycrystalline perovskites, the low Q-factor of lasers could lead to a loss in the coupling propagation.
- (2) Reference of *Adv. Mater.* 29, 1604268 (2017) demonstrates the wavelength of reflection resonance could be tuned by perovskite metasurfaces. However, the Q-factor of perovskite metasurfaces of 2-5 is low, which cannot support the lasing emission, in-plane coherent propagation and waveguide coupling.
- (3) Reference of *Nat. Photonics* 14, 82 (2019) demonstrates the in-situ synthesis of perovskite QDs inside glass by 3D laser printing to form fluorescence micro patterns. Indeed, this work is the pioneer in 3D laser printing fabrication for perovskite. Because the ablation process is difficult to generate lasing, the emissions of such micro patterns are fluorescence not lasing (i.e., coherent light), which could not carry out waveguide coupling and coherent propagation. The propagation length of fluorescence is limited and shorter.

Compared with these previous works, we would like to highlight the novelty, differentiation, and importance of our work in the following points:

- (1) More importantly and intriguingly, **we first-time homogeneously integrated with microlasers and multiple functional micro photonic devices** (*i.e.*, micro beam splitter, X-coupler, Mach-Zander interferometer) **to realize the coupled propagation and manipulation for on-chip coherent light** (split and coalescence) **in perovskite systems, which could simply demonstrate the basic transfer matrix of calculation in photonic chips** [*Phy. Rev. Lett.* 73, 58 (1994); *Nat. Photonics* 11, 441 (2017)]. This study provides novel alternative paths toward integrated photonic chips, integrated optical quantum devices, and photonic calculations [*Nat. Photonics* 605, 457 (2022)].
- (2) Silicon-based integrated photonic devices as passive systems need to be heterogeneously integrated with active light sources, which is difficult in technology. **Thus, active integrated photonic circuits, based on perovskite with great optoelectronic performance and simple fabrication process, play an important role in future integrated photonics research.**
- (3) Compared with electronic and optoelectronic devices, photonic and all-optical devices possess ultrafast propagation speed, low energy consumption, and rich modulation means, which is conducive to the development of photonic chips and optical communications.

To highlight the novelty, differentiation, and importance of our work, we have added the following words in the main text.

“Essential functions of macroscopic optical circuits, such as signal generation, directional propagation, directional beam splitting, transfer matrix, and phase modulation, can be achieved through corresponding integrated photonic devices. Inspired by this idea, perovskite-based active integrated photonic devices were designed and fabrication, including micro lasers, waveguide couplers, micro beam splitters and integrated interferometers, to demonstrate basic functionalities of optical matrix operation (allocation, transfer and modulation), as shown in schematic formulas of Fig. 1a-c and details in Supplementary Materials.”

Comments 3:

The PL spectrum shown in Figure 1h is not symmetric. Why? It seems that the perovskite microstructures showed significant amount of trap states near the band edge, and the PL spectrum showed a tail in the long-wavelength region.

Response 3:

We thank the reviewer for pointing out this issue. We agree with the comments of the reviewer about the asymmetry of the photoluminescence (PL) emission spectrum for the monocrystalline CsPbBr₃ thin film shown in Figure 1h, due to the existence of surface defect states. Therefore, we optimized the growth process of the film to reduce the defect state, then re-characterized the PL emission of the thin film and obtained a relatively symmetrical PL spectrum, as shown in Fig. R1.

In order to further verify that the PL emission of the thin film contains a small number of defect states, we also re-measured the time-resolved PL (TRPL) spectrum of the film, which will be shown in detail in the next comment.

Although the spontaneous emission in CsPbBr₃ thin film contains a small number of components for defects or trap state emission, our active integrated photonic devices operate in the lasing region (coherent emission) and then utilize such generated microlaser to realize the manipulation of waveguide coupling and propagation. Therefore, a small number of trap-state emissions has little impact on the function of the devices and the concept we proposed.

Figure R1. Room-temperature PL emission spectrum of the monocrystalline CsPbBr₃ thin film.

In order to improve the quality of the manuscript, we have replaced the PL emission of the thin film in Figure 1h with a more symmetrical one, and the corresponding discussion in the main text has also been revised as:

“The normalized PL and absorption spectra in Fig. 1h show an exciton emission peak centered at 525.2 nm with a linewidth of 14.4 nm and a strong excitonic absorption centered at 515.4 nm.”

Comments 4:

The TRPL lifetime is too short compared with high-quality CsPbBr₃ single crystals. The lifetime of ~5 ns indicates a fast recombination process, which could be caused by the trap states as we observed from the PL spectrum. In addition, the authors cannot attribute the two lifetimes to two different recombination processes and cited two papers. The physical meaning is not sound. There are also many reports on CsPbBr₃ perovskites, which showed mono-exponential decays.

Response 4:

We thank the reviewer for pointing out this issue. Following the comments of the reviewer, we re-characterized and analyzed the TRPL spectra for delay dynamics of the monocrystalline CsPbBr₃ thin film, as shown in Fig. R2.

Figure R2. Time-resolved PL spectrum of the monocrystalline CsPbBr₃ thin film, the fitting decay profile giving two lifetime components of 12.74 ns and 96.52 ns.

Two different lifetime components of thin film, *i.e.*, a fast component of 12.74 ns and a slow component of 96.52 ns, can be fitted by an exponential decay function, as shown in Fig. R2. Such two different time scales can be attributed to the bulk emission recombination (slow) together with the presence of surface trap states (fast), indicating that carriers can propagate deep in the material [*Science* 347, 519 (2015); *Adv. Mater.* 30, 1705992 (2018); *Adv. Opt. Mater.* 5, 1600704 (2017)]. This result agrees with the values reported in previous literature [*Adv. Mater.* 30, 1705992 (2018); *Nano Lett.* 21, 1454 (2021); *Adv. Opt. Mater.* 5, 1600704 (2017)], which suggests the high optical quality of the CsPbBr₃ thin film.

To strengthen the quality of our paper, we have replaced the TRPL spectrum in Figure 1i and given a more explicit description of the TRPL decay dynamics in the main text, as follows:

“TRPL delay dynamics (Fig. 1i) reveals two different lifetime components of the thin film, *i.e.*, a fast component of 12.74 ns and a slow component of 96.52 ns. Such two different time scales could be attributed to the radiative recombination of exciton (slow) together with the presence of surface trap states (fast)^{34,41,42}.”

To clarify this mechanism, we have added the corresponding references as Ref. 41 and 42.

[41] Shi, D. et al. Low trap-state density and long carrier diffusion in organolead trihalide perovskite single crystals. *Science* **347**, 519-522 (2015).

[42] Saidaminov, M. et al. Inorganic Lead Halide Perovskite Single Crystals: Phase-Selective Low-Temperature Growth, Carrier Transport Properties, and Self-Powered Photodetection. *Adv. Opt. Mater.* **5**, 1600704 (2017).

Comments 5:

The authors should provide the PLQY value of these samples to show the quality of the obtained CsPbBr₃ microstructures. Ideally, the light-intensity dependent PLQY should be presented for the samples with and without patterning.

Response 5:

We thank the reviewer for his/her suggestion. Following the comments of the reviewer, we measured the photoluminescence quantum yield (PLQY) of the monocrystalline CsPbBr₃ thin film and the etched microwire under low exciting power, and then obtained the values for thin film and microwire of 10.49% and 10.25%, respectively. This result indicates that the PLQY of CsPbBr₃ obtained microstructures has remained relatively stable after focused ion beam (FIB) treatment.

Due to the micrometer scale of our microstructures, we need to carry out PLQY experiments on a home-built micro-PL spectrum configuration at room temperature (Fig. R3). Figure R3a displays the spectra of the excitation laser passing through transparent mica (reference) and CsPbBr₃ thin film, measured by a 50× objective (NA = 0.75). The absorption of thin film can be obtained by the area difference between the two laser peaks. Figure R3b shows the corresponding PL emission spectrum of this perovskite thin film excited by the same laser with the same power.

Figure R3. (a) Room-temperature spectra of exciting laser through bare mica (red curve) and monocrystalline CsPbBr₃ thin film (blue curve), exposure time: 0.1 s. The inset is the magnified peaks of the exciting laser for clarifying the absorption. (b) PL spectrum of the perovskite thin film excited by the same laser with the same power corresponding to (a), exposure time: 30 s. (c) Spectra of exciting laser through bare mica and CsPbBr₃ microwire, exposure time: 0.1 s. (d) PL spectrum of the perovskite thin film excited by the laser with the same power corresponding to (c), exposure time: 30 s.

Thus, PLQY of the perovskite thin film can be described by the following equation:

$$\text{PLQY} = \frac{I_{\text{Emission}}}{\alpha \cdot I_{\text{Absorption}}} \quad (1)$$

where I_{Emission} is the area of the PL emission peak for the perovskite thin film, $I_{\text{Absorption}}$ is the area difference between the two laser peaks, and α is the spatial volume ratio of the solid angle for collecting PL emission signal which is related to the numerical aperture of the objective. α can be calculated as 0.1693 by sphere integral. Finally, an approximate PLQY of the perovskite thin film under low exciting power can be calculated as 10.49%. Utilizing a similar

characterization method, the PLQY of the perovskite microwire is measured and calculated as 10.25%, showing a slight variation after FIB treatment (Fig. R3c and d).

While PLQY is an essential parameter for optoelectronic devices such as LEDs and solar cells, in photonic and all-optical devices (*i.e.*, our work), the function and concept of our proposed active integrated photonic devices pay more emphasis on the capabilities of propagation and coupling for the laser signals, and the manipulation for coherent light on the chip.

Comments 6:

The authors discussed and designed the optical cavity. Hence, a cavity geometry dependent device performance should be provided. I mean, the authors should give a full comparison how the cavity geometry affects the light emission.

Response 6:

We thank the reviewer's comment. Following the reviewer's suggestion, we have measured the lasing properties of the multiple different length and width CsPbBr₃ microwires, to investigate the cavity geometry-dependent lasing performance, as shown in Fig. R4 and R5. Moreover, an etched CsPbBr₃ microdisk with whispering gallery mode (WGM) for lasing has been still investigated as a comparison, as shown in Fig. R6.

Figure R4a displays the top view scanning electron microscopy (SEM) images of five etched different-length CsPbBr₃ microwires, with the dimensions of 6 μm × 3 μm, 8 μm × 3 μm, 10 μm × 3 μm, 15 μm × 3 μm and 20 μm × 3 μm, respectively. All the microwires exhibit smooth surfaces and sharp edges with an identical thickness of 457 nm (Fig. R4d). Under the excitation by a 400 nm femtosecond pulsed laser above the threshold, the PL microscope images reveal that strong lasing emissions distinctly leak out from the opposite end facets of the microwires, which can be attributable to the Fabry-Pérot (FP) mode oscillation (Fig. R4b). The normalized lasing emission spectra for the five different-length microwires are shown in Fig. R4c, demonstrating the evolution of the lasing mode number *versus* the length. The lasing mode

numbers increase from 2 to 6 when the lengths of microwires change from 6 μm to 20 μm (Fig. R4e). Thus, we can precisely control the number of output lasing modes by adjusting the length of the FP microcavity. To compare the change of lasing threshold, we set the threshold for microwire with the dimension of 6 $\mu\text{m} \times 3 \mu\text{m}$ as P_0 , and calculated the threshold ratio of these different-length microwires. One can see that lasing thresholds of perovskite microwires exhibit a decay trend with the length increasing, which could be attributed to the enlarged gain medium region (Fig. R4f).

Figure R4. (a) SEM images of CsPbBr₃ microwires with identical widths of 3 μm and different lengths from 6 μm to 20 μm . Scale bar: 4 μm . (b) Corresponding PL microscope images of the different-length microwires above the threshold. Scale bar: 4 μm . (c) The normalized lasing emission spectra of the different-length microwires above the threshold. (d) SEM image of the side facet for one microwire (tilted by 50° with respect to the horizontal position), indicating a height of 457 nm. Scale bar: 300 nm. Such microwires are of identical thickness. (e) Evolution of the lasing mode number as a function of the length of microwire. (f) Evolution of the lasing threshold as a function of the length of microwire, setting the threshold of microwire with the dimension of 6 $\mu\text{m} \times 3 \mu\text{m}$ as P_0 .

In the same method, the lasing properties of the etched different-width CsPbBr₃ microwires are investigated. Figure R5a shows the top view SEM images of five different-width microwires, with the dimensions of 15 μm \times 0.5 μm , 15 μm \times 1 μm , 15 μm \times 2 μm , 15 μm \times 3 μm and 15 μm \times 4 μm , respectively. All the microwires possess smooth surfaces and sharp edges with an identical thickness of 444 nm (Fig. R5d). Excited by a 400 nm femtosecond pulsed laser above the threshold, one can see that such microwires display strong lasing emissions in the same type, which distinctly leak out from the opposite end facets and originate from the FP oscillation modes (Fig. R5b). Figure R5c displays the normalized lasing emission spectra for the five different-width microwires, demonstrating the evolution of the lasing mode number *versus* the width. Unlike the results of the different-length microwires, due to the length limitation of the FP microcavity, the number of lasing modes is stable around 5 with the change of width (Fig. R5e). As for the variation trend of the lasing threshold, similarly, we set the threshold for microwire with the dimension of 15 μm \times 0.5 μm as P₁, and calculated the threshold ratio of these different-width microwires. The results also indicate that the thresholds of the perovskite microwires exhibit a decay trend with the width increasing, attributed to the enlarged gain medium region (Fig. R5f).

Figure R5. (a) SEM images of CsPbBr₃ microwires with identical lengths of 15 μm and different widths from 0.5 μm to 4 μm. Scale bar: 4 μm. (b) Corresponding PL microscope images of the different-width microwires above the threshold. Scale bar: 4 μm. (c) The normalized lasing emission spectra of the different-width microwires above the threshold. (d) SEM image of the side facet for one microwire (tilted by 50° with respect to the horizontal position), indicating a height of 444 nm. Scale bar: 300 nm. Such microwires are of identical thickness. (e) Evolution of the lasing mode number as a function of the width of microwire. (f) Evolution of the lasing threshold as a function of the width of microwire setting the threshold of microwire with the dimension of 15 μm × 0.5 μm as P₁.

Meanwhile, we measured the lasing property of an etched CsPbBr₃ microdisk to compare the effect of the cavity geometry on lasing oscillation modes. Figure R6a shows the top view SEM image of the obtained CsPbBr₃ microdisk after FIB treatment, where the disk has a smooth surface with a diameter of 8 μm. Also, the smooth surface and sharp edges of the microdisk can be verified from the magnified SEM image, and the thickness is extracted as 466 nm (Fig. R6b). At room temperature, under low-power excitation by a 400 nm femtosecond pulsed laser, the whole disk presents a near-uniform PL emission (Fig. R6c). Under high pump fluence and

above the threshold, strong PL emission appears around the disk, which is visibly distinguished from emission in the in-plane region and indicates the WGM mode laser arises (Fig. R6d).

Figure R6. (a) SEM image of a CsPbBr₃ microdisk with a diameter of 8 μm after FIB treatment. Scale bar: 4 μm. (b) SEM image of the side facet for the microdisk (tilted by 50° with respect to the horizontal position), indicating a height of 466 nm. Scale bar: 400 nm. (c-d) Corresponding real-space PL images of the microdisk below (c) and above (d) the threshold, respectively. Scale bar: 4 μm. (e) PL spectra emitted from the microdisk with the pump fluence increasing from 51 μJ cm⁻² to 66 μJ cm⁻². (f) Evolution of the integrated PL emission intensity (red curve) and linewidth (blue curve) as functions of pump fluence of the microdisk, showing the threshold of 53.8 μJ cm⁻². (g) One magnified lasing oscillation mode with a Lorentz-fitted linewidth of 0.198 nm and a Q-factor of 2718.

Figure R6e displays the PL emission evolution with the pump fluence of the microdisk. One can see that with the pump fluence increasing from 51 μJ cm⁻² to 66 μJ cm⁻², several sharp peaks of lasing emission emerge at the low-energy side of spontaneous emission (SE), and become dominant in the PL spectra with the intensity rapidly rise. A nonlinear process of the

lasing emission in the microdisk is revealed by the evolution of the integrated PL intensity and linewidth as functions of pump fluence (Fig. R6f). The dramatical decrease of the linewidth and the typical S-shaped growth curve of the PL intensity unambiguously indicate the arising of lasing behavior with the pump fluence increasing. Here, the critical threshold of the microdisk is extracted as $53.8 \mu\text{J cm}^{-2}$. Figure R6g illustrates the Lorentz fitting of one magnified lasing oscillation mode with a linewidth of 0.198 nm and a Q-factor of 2718 at $1.12 P_{\text{th}}$, indicating the high lasing performance of the etched perovskite microdisk.

These results demonstrate that the lasing performance of the etched CsPbBr₃ laser can be precisely controlled by adjusting the geometry of the microcavity. Thus, a high-quality micro laser can be achieved through in-situ FIB etching on demand.

To strengthen the quality of our paper, we have added the discussion about the cavity geometry-dependent lasing performance of the etched perovskite micro laser in Support Information, and given some explanation in the main text, as follows:

“To further investigate the cavity geometry-dependent lasing performance, we measured the lasing properties of the multiple different length and width microwires as well as an etched microdisk (Fig. S5-S7). These results demonstrate that the lasing performance of the etched CsPbBr₃ laser can be precisely controlled by tuning the geometry of the microcavity. Thus, a high-quality, coherent and stable microwire laser can be achieved through in-situ FIB etching on demand (Fig. S8).”

Comments 7:

The FDTD simulation of the 2D normalized electric field intensity distribution along the microwire is meaningless. Any light emitting medium in the microwire should result similar pattern. It will be more helpful, if the authors could give more information on how to design a decent cavity, and discuss how other parameters affect the device performance. For instance, the charge carrier lifetime, the film surface roughness and etc.

Response 7:

We thank the reviewer for pointing out this issue. According to the comments of the reviewer, we give a more specific description of the design of a decent FP microcavity, and discuss the influence of the surface roughness of FP microcavity on the lasing performance, as shown in Fig. R7.

The result of the FDTD simulation in Figure 2d aims to explain that the FP cavity consists of both ends of the CsPbBr₃ microwire and allows an effective in-plane laser output above the threshold. Thus, the etched microwire could serve as the coupled lasing source for the integrated photonic devices mentioned later in the manuscript.

The output signal of a photonic device is a light intensity signal that averages multiple pulse times, which belongs to the steady electromagnetic field distribution. As a consequence, the effect of carrier lifetime on the photonic devices here can be ignored.

In a realistic situation, the Q-factor of isolated microcavity could be attributed to three factors

$$Q^{-1} = Q_c^{-1} + Q_a^{-1} + Q_s^{-1} \quad (2)$$

where the first term $1/Q_c$ corresponds to the intrinsic photon leakage associated with the imperfect reflection resulting from the boundaries of the microcavity, $1/Q_a$ accounts for the absorption loss of the material, and $1/Q_s$ describes the loss due to light scattering from surface roughness [*J. Am. Chem. Soc.* 137, 9289 (2015)]. Obviously, the quality and performance of FP microcavities can be affected by their surface roughness and shape morphology, *e.g.*, it is reported that single-crystal MAPbBr₃ microcavity having sidewall roughness less than 5 nm support lasing with Q-factor up to 6000 [*Adv. Mater.* 29, 1606205 (2017)], whereas polycrystalline MAPbI₃ cavities with rougher sidewall (> 8 nm) fabricated through a mask yield lasing with Q-factor up to 710 [*Nano Lett.* 18, 6915 (2018)].

In our experiment, the etched CsPbBr₃ microlasers process regular structures, smooth surfaces and sharp edges (details in Support Information). The lower optical losses and light scattering losses result from the nearly ideal platform for internal total reflection (the first term $1/Q_c$).

Furthermore, the low self-absorption around the lasing region is confirmed by the room temperature absorption spectra in Figure 1h (the second term $1/Q_a$). Thus, one can observe that a high-quality CsPbBr₃ microcavity is mainly attributed to its regular shape morphology and smooth surface. The atomic force microscopy (AFM) displays the height profile of the end facet for a CsPbBr₃ microwire in Fig. R7a and shows the sharp edge and lower roughness of the cavity after FIB treatment. This meets the requirement of high-quality microcavity, corresponding to the lasing oscillation mode with a Q-factor of 2460 in Figure 2e.

Meanwhile, we simulated the influence of the surface roughness of FP microcavity on the lasing output direction by FDTD Solutions. As the end facet of the microwire changes from the ideal smooth side to a rough side with a fluctuation of 20 nm, the direction of the lasing emission for CsPbBr₃ microwires undergoes a great change, as shown in Fig. R7b and c. Therefore, **the larger surface roughness of the microwire laser will affect the lasing propagation direction and coupling efficiency of photonic devices.**

Figure R7. (a) Atomic force microscopy height profile of the end facet for a CsPbBr₃ microwire after FIB treatment, showing a typical thickness of 300 nm and a sharp edge. (b-c) FDTD simulation of the 2D normalized electric field intensity distribution around the end facet of the microwire ($\lambda = 537.5$ nm, $n = 2.53$), with an ideal smooth side for (b) and a rough side with fluctuation of 20 nm for (c), respectively. The white dashed lines depict the region of the end facet for the microwire.

Comments 8:

The device performance should be compared with other perovskite micro-cavities and other conventional semiconductors, ideally in a table.

Response 8:

We thank the reviewer's comment. In our work, through the in-situ FIB etching process, the achieved high-quality perovskite micro laser can be on-demand and homogeneously integrated with multiple functional micro photonic devices. A significant advantage is that in-situ fabrication, cavity geometry and lasing properties can be controlled, which provides a novel avenue toward high-efficiency active light source for photonic devices. In recent years, while different kinds of emerging semiconductor materials have shown attractive lasing performance, few studies have reported their applications for functional integrated photonic devices on chips.

Following the reviewer's suggestion, we compared the lasing performance of the etched CsPbBr₃ microwire with other perovskite microstructures and other conventional semiconductors in Table R1. It is encouraging that our work realizes an efficient lasing output with a high Q-factor, based on in-situ pattern processing at arbitrary position.

Table R1. Comparisons of Main Laser Parameters among Reported Semiconductor Nano/Microlasers^a.

Material & Structure	Cavity Type	Position	Q-Factor	Threshold ($\mu\text{J cm}^{-2}$)	Reference
MAPbI ₃ NPL	WGM	Random	650	37	1
MAPbX ₃ NW	FP	Random	3600	0.22	2
CsPbX ₃ MS	WGM	Random	6100	0.42	3
CsPbX ₃ MD	WGM	In-situ	530	10.3	4
MAPbI ₃ Film	VCSEL	In-situ	1100	7.6	5
CsPbBr _x I _{3-x} NW	FP	Random	1736	18	6
ZnO NW	FP	Random	485	400	7
GaN NP	PC	In-situ	170	40000	8

ZnSe NW	FP	Random	640	340	9
CdSe NB	FP	Random	3962	18	10
CsPbBr ₃ MW	FP	In-situ	2460	48.7	Our work
CsPbBr ₃ MD	WGM	In-situ	2718	53.8	Our work

^aMA, NPL, NW, MS, MD, NP, NB, MW, WGM, FP, VSCEL and PC denote CH₃NH₃, nanoplatelet, nanowire, microsphere, microdisk, nanopillar, nanobelt, microwire, whispering gallery mode, Fabry-Pérot, vertical-cavity surface-emitting laser and photonic crystals, respectively.

Reference: [1] *Nano Lett.* 14, 5995 (2014); [2] *Nat. Mater.* 14, 636 (2015); [3] *ACS Nano* 11, 10681 (2017); [4] *Adv. Mater.* 29, 1604510 (2017); [5] *Adv. Mater.* 29, 1604781 (2017); [6] *Adv. Mater.* 30, 1800596 (2018); [7] *J. Am. Chem. Soc.* 131, 2125 (2009); [8] *IEEE J. Sel. Top. Quantum Electron.* 19, 4900206 (2013); [9] *Adv. Optical Mater.* 1, 319 (2013); [10] *Appl. Phys. Lett.* 110, 201112 (2017).

Comments 9:

I am not sure if it is appropriate to term the devices as “laser”. Some of the papers in the field also term them as “amplified spontaneous emission (ASE)”. The observed small line width of these perovskite samples under high light excitation showed multiple modes, and there are also many emitted photons showed different energies. It seems that they are not proper laser devices at current stage.

Response 9:

To address the concern of the reviewer, we have performed new experiments for the evolution of PL spectra for a CsPbBr₃ microwire by the pump fluence increasing (Fig. R8), which unambiguously reveals a typical and complete process of the nonlinear transition from SE via amplified spontaneous emission (ASE) to stimulated radiation (*i.e.*, lasing). The nonlinear

transition of the PL emission intensity of the CsPbBr₃ microwire can be demonstrated by the power-dependent profile of PL intensity and angle-resolved PL (ARPL) interference patterns.

Typically, the most significant difference among SE, ASE and lasing is the linewidth (*i.e.*, SE: 10-100nm; ASE: 4-10nm; lasing: < 1 nm) and spatial coherence. With the pump power increasing, SE, ASE and lasing appear in the order, and their order in wavelength is SE, ASE and lasing, moving towards the longer wavelength. Regarding spatial coherence, SE and ASE are incoherent, lasing emission process a good coherence.

Figure R8. Evolution of PL spectra for a CsPbBr₃ microwire with the pump fluence increasing, unambiguously revealing the complete process of the transition from spontaneous emission *via* amplified spontaneous emission to stimulated radiation (*i.e.*, lasing).

ASE is spectrally different from PL emission, with a narrower bandwidth of 4-9 nm and a red-shifted for the SE maximum [*Nat. Commun.* 6, 8056 (2015)]. With the pump fluence increasing, the lasing behavior arises from the unique ability of such microwire to act as both gain medium and laser microcavity. Upon reaching a sufficient carrier density, **stimulated emission takes over and some sharp peaks of lasing emission emerge on the top of the ASE peak in the**

PL spectrum. This typical phenomenon for our microwire laser is shown in Fig. R8, where the linewidth of SE, ASE and lasing are extracted as 15.2 nm, 7.5 nm and 0.53 nm, respectively.

The other difference between ASE and lasing is spatial coherence. The spatial coherence of the microwire lasing can be verified by the ARPL measurement. Above the threshold, the ARPL spectrum of the microwire unambiguously demonstrates standing wave-like patterns extending to all detection angles, which originates from the interference between coherent lasing modes emitted from the opposite end facets (Figure 2g).

Reply to Reviewer #2

Comments 1:

The manuscript entitled “Inorganic Perovskite-Based Active Multifunctional Integrated Photonic Devices”, presents experimental demonstrations of active multifunctional integrated photonic devices based on all-inorganic perovskite, including microlasers, micro beam splitters, X-couplers, and Mach-Zehnder interferometers. Overall, this study is likely to be of interest to the broad readership of Nature Communication and contribute to the general research community in this field. However, several issues should be addressed by the authors prior to recommending the manuscript for the publication stage. Additional technical comments are as follows.

Response 1:

We would like to thank reviewer #2 for his/her careful review and positive comments on our manuscript. Reviewer #2 provided constructive suggestions for improving our paper, and we have carefully accommodated his/her comments and performed further experiments in the revised manuscript, which will be discussed below. Thus, we believe that the revised manuscript is more suitable for publication in *Nature Communications*.

Comments 2:

The authors chose mica as substrates for the deposition of perovskite thin films, the effect of the substrate on the crystal quality of the perovskite thin films should be discussed.

Response 2:

We thank the reviewer’s comment. Lattice match with substrate is crucial for the growth of large area single crystal perovskite thin films. In our work, the monocrystalline CsPbBr₃ thin

film as the raw material of photonic devices is grown by a chemical vapor deposition (CVD) process on mica. The synthesized thin film exhibits high optical quality and large areas, which meets the requirements of device fabrication. This epitaxial growth model here can be understood as: CsPbBr₃ expands along [100] and [010] directions that are paralleled to [100] and [010] directions of the mica, respectively [*Nano Lett.* 21, 1454 (2021)]. Considering the lattice spacings of cubic phase CsPbBr₃ and mica: $d_{\text{CsPbBr}_3(100)} = 5.830 \text{ \AA}$, $d_{\text{CsPbBr}_3(010)} = 8.245 \text{ \AA}$ (JCPDS no. 45-0752), $d_{\text{mica}(100)} = 5.387 \text{ \AA}$, and $d_{\text{mica}(010)} = 9.324 \text{ \AA}$ (JCPDS no. 16-0344), calculation suggests that: along CsPbBr₃ [010] direction, 8 periodicities of CsPbBr₃ [010] match with 5 periodicities of mica [010], with a 0.04% mismatch, while long CsPbBr₃ [100] direction, 8 periodicities of CsPbBr₃ [100] match with 9 periodicities of muscovite mica [100], with a 3.95% lattice mismatch. Thus, such imperceptible lattice mismatch meets the requirement of high quality and large areas of CsPbBr₃ thin film synthesized on mica.

Meanwhile, to compare the effect of the substrate on the crystal quality of the perovskite material, we also synthesized CsPbBr₃ by a CVD process on other substrates, i.e., Si, GaN, Sapphire, and ITO, as shown in Fig. R9. The scanning electron microscopy (SEM) images of CsPbBr₃ microplates synthesized on Si substrate reveal the highly smooth surfaces and ideal thickness for confining photons of the microstructures (Fig. R9a and b). Uniform fluorescence emission also demonstrates the high crystal and optical quality (Fig. R9c). However, the asymmetric and large lattice mismatch along different crystallographic directions between CsPbBr₃ lattice and Si substrate, results in the formation of the single microplate morphology [*ACS Photonics* 7, 454 (2020); *ACS Nano* 11, 1189 (2017)]. Due to such an epitaxial growth model, similar microplate structures can be achieved on GaN, Sapphire, and ITO substrates (Fig. R9d-f, [*ACS Nano* 13, 10085 (2019); *ACS Photonics* 9, 2431 (2022)]).

Figure R9. (a-b) SEM images of CsPbBr₃ microplates synthesized on Si substrate. Scale bars: 15 μm (a) and 20 μm (b). (c-f) Optical microscopy images of CsPbBr₃ microplates synthesized on Si (c), GaN (d), Sapphire (e), and ITO (f) substrates, respectively. The insets are corresponding fluorescence microscope images. Scale bars: 15 μm (c), 10 μm (d), 10 μm (e) and 20 μm (f).

Comments 3:

Since the crystal quality of the perovskite thin films is very important for the properties of the devices, a more comprehensive analysis of the surface elemental composition and chemical state of the monocrystalline perovskite thin films is recommended.

Response 3:

We thank the reviewer's comment. Following the reviewer's suggestion, we performed X-ray photoelectron spectroscopy (XPS) measurement on monocrystalline CsPbBr₃ thin film. The XPS peaks of single Cs_{3d}, Pb_{4f}, and Br_{3d} are collected as shown in Fig. R10. Two peaks at around 738.2 eV and 724.3 eV can be assigned to Cs 3d_{3/2} and Cs 3d_{5/2}, while the strong peaks located at 143.0 eV and 138.1 eV are attributed to Pb 4f_{5/2} and Pb 4f_{7/2}, respectively. The prominent peaks of Br 3d_{3/2} and Br 3d_{5/2} of Br_{3d} are observed at around 69.1 eV and 68.0 eV. This result is consistent with the values reported in previous literature [*Adv. Mater.* 29, 1701636 (2017); *Adv. Opt. Mater.* 6, 1800879 (2018)]. Meanwhile, analysis of the presence of Cs, Pb,

and Br elements is nearly consistent with the material composition of CsPbBr₃, demonstrating the high chemical purity of the CsPbBr₃ thin film on mica through direct CVD synthesis.

Figure R10. (a-c) XPS analysis of the monocrystalline CsPbBr₃ thin film for Cs_{3d} (a), Pb_{4f} (b), and Br_{3d} (c), respectively. The presence of Cs, Pb and Br elements is consistent with the material composition of CsPbBr₃.

To strengthen the quality of our paper, we have added the XPS analysis of the monocrystalline CsPbBr₃ thin film in Support Information, and given some explanation in the main text, as follows:

“X-ray photoelectron spectroscopy collected the signals of single Cs_{3d}, Pb_{4f}, and Br_{3d} to confirm the chemical states and chemical purity of the thin film (Fig. S3).”

Comments 4:

The high energy of the electron beam during FIB may have some effect or even damage on the perovskite films, the authors should discuss the effect of FIB on the PL lifetimes of perovskite, i.e., including PL results before and after FIB etching process.

Response 4:

We thank the reviewer’s comment. We agree with the reviewer’s comment that perovskite materials will be damaged under the irradiation of electron and ion beams [*ACS Appl. Mater.*

Interfaces 11, 15756 (2019)]. Following the suggestion, in order to assess the extent of the damage caused by focused ion beam (FIB) etching, we have performed new experiments to characterize the photoluminescence (PL) emission and time-resolved PL (TRPL) delay dynamics of the monocrystalline CsPbBr₃ thin film and microstructure (after FIB treatment), respectively.

The nearly consistent shape and strength of the PL emission before and after FIB treatment demonstrate the feasibility of fabricating photonic devices directly (Fig. R11a and c). As for the TRPL lifetime, one can observe the two shorter different time scales which are attributed to the bulk emission recombination (slow) together with the presence of surface trap states (fast) after FIB treatment, indicating more carrier trap defects are formed under higher ion-dose irradiation (Fig. R11b and d, [*Science 347, 519 (2015)*]).

It should be noted that our active integrated photonic devices operate in the lasing region (coherent emission), and then utilize such generated microlaser to realize the manipulation of waveguide coupling and propagation. Therefore, such a small number of carrier trap defects has little impact on the function of the devices and the concept we proposed.

Figure R11. (a-d) Room-temperature PL emission and Time-resolved PL delay dynamics contrast of the monocrystalline CsPbBr₃ thin film ((a) and (b), before FIB treatment) and microstructure ((c) and (d), after FIB treatment), showing the feasibility of directly fabricating photonic devices.

To strengthen the quality of our paper, we have replaced and added the PL emission and TRPL delay dynamics contrast of the thin film before and after FIB treatment in Support Information, and revised the words in the main text, as follows:

“Furthermore, the intensity and shape of photoluminescence (PL) emission for the perovskite thin film remain nearly consistent before and after FIB treatment (Fig. S1 of Supplementary Materials), thus substantiating the feasibility of directly fabricating photonic devices.”.

Comments 5:

Since the stability is always a major concern for halide perovskite materials, it would be informative to investigate whether high pump excitations of microwire laser will change the lasing properties of the microlasers, such as the lasing threshold, and peak wavelength.

Response 5:

We are grateful to Reviewer 2 for the valuable comments. The optical stability of microwire lasers is crucial for the efficient operation of integrated photonic devices. According to the reviewer’s suggestions, we measured the lasing properties of five CsPbBr₃ microwires of the same size to investigate the stability and uniformity of the designed laser devices.

As shown in Fig. R12a and b, a typical etched CsPbBr₃ microwire exhibits a smooth surface and sharp edges with a dimension of 10 μm × 2 μm × 0.444 μm. Under low-power excitation by a 400 nm femtosecond pulsed laser, the PL microscope image of the microwire reveals a uniform green-color emission (Fig. R12c). When the pump fluence increases above the threshold, strong lasing emission is observed to distinctly leak out from the opposite end facets

of the microwire, owing to the Fabry-Pérot (FP) mode oscillation (Fig. R12d). The normalized lasing emission spectra for the five microwires of identical dimensions are shown in Fig. R12e, demonstrating the near-unanimous peak shape and number of lasing modes. One can observe that under the same measurement configuration, the thresholds of these microwire lasers exhibit small fluctuation, which also indicates the uniformity of the designed laser devices (Fig. R12f).

Moreover, we performed measurements for the stability and robustness of microwire laser. The emission intensity as a function of pump fluence is measured a few days after the first measurement. One can observe that the emission intensity and threshold of lasing have hardly changed much, as shown in Fig. R12g. Furthermore, we measured the power-dependent emission intensity by repeatedly increasing and decreasing pumping power, and we still observed that the lasing did not degenerate, as shown in Fig. R12h. Thus, we believe the etched perovskite microwire laser is stable and robust under high optical excitation.

Figure R12. (a) SEM image of a CsPbBr₃ microwire with a dimension of 10 μm × 2 μm. Scale bar: 3 μm. (b) SEM image of the side facet for the microwire (tilted by 50° with respect to the horizontal position), indicating a height of 444 nm. Scale bar: 300 nm. (c, d) Corresponding PL microscope images of the microwire below (c) and above (d) the threshold, respectively. Scale bar: 3 μm. (e) Normalized lasing emission spectra for five CsPbBr₃ microwires of identical dimension, showing near-unanimous peak shape and number of lasing modes. (f) Statistics of the lasing threshold for the five microwires. (g, h) Integral emission intensity as a function of pump fluence under different times and excitation sequences.

To strengthen the quality of our paper, we have added the analysis of the stability and uniformity of etched CsPbBr₃ perovskite microwire laser in Support Information, and revised the words in the main text, as follows:

“Thus, a high-quality, coherent and stable microwire laser can be achieved through in-situ FIB etching on demand (Fig. S8).”.

Comments 6:

It is nice that the authors have demonstrated a Mach-Zehnder interferometer (MZI) integrated with a microwire lasing source, the potential modulation properties of this MZI modulator prototype for future applications should be discussed. In addition, the microwire laser and the etched MZI are coupled via a gap of 122.5 nm, the effect of this gap should also be discussed.

Response 6:

We thank the reviewer’s comment. Following the reviewer’s comment, we give a brief description about the potential modulation properties of such perovskite-based Mach-Zehnder interferometer (MZI) modulator prototype for future applications as follows:

One of the modulation potentials in future applications of MZI devices is the introduction of an electrical structure, *e.g.*, applying a voltage to one arm of the device, to change the refractive index of materials and optical length, which can be used to modulate the phase and intensity of

light propagation. After the coalescence of the beams from the two arms, the phase change caused by the electrical structure modulation will be transformed into the light intensity change of the output light signals, to realize the signal modulation at the output end. In future work, various hybrid optoelectronic structures can be integrated with the perovskite-based active MZI to broaden its applications.

Following the reviewer's suggestion, to investigate the effect of the coupling gap on the coupled propagation efficiency for lasing, we performed lasing waveguide-coupling measurements between two etched CsPbBr₃ microwires separated by different coupling gaps, as shown in Fig. R13. The result displays a decay trend of the propagation efficiency for lasing with the gap distance increasing.

Figure R13. (a) SEM images of CsPbBr₃ waveguide couplers with different gaps between two microwires. The microwires are of identical dimension (10 μm × 2 μm). Scale bar: 5 μm. (b-f) Magnified SEM images of the different gaps between two microwires, showing the gaps of 126 nm (b), 211 nm (c), 271 nm (d), 356 nm (e) and 533 nm (f). Scale bar: 500 nm. (g) SEM image of the side facet for one waveguide coupler (tilted by 50° with respect to the horizontal position), indicating a height of 548 nm. Scale bar: 500 nm. Such waveguide couplers are of

identical thickness. **(h)** Lasing microscope images of the waveguide couplers with different gaps under the pump fluence of $62 \mu\text{J cm}^{-2}$. Scale bar: $5 \mu\text{m}$. The white dashed circles and boxes depict excitation terminals and propagation terminals, respectively. **(i)** Evolution of coupled propagation efficiency for lasing as a function of the gap distance of microwire waveguide coupler, displaying a decay trend with the gap distance increasing.

The CsPbBr_3 waveguide couplers with different coupling gaps are composed of two microwires with identical dimensions of $10 \mu\text{m} \times 2 \mu\text{m} \times 0.548 \mu\text{m}$ (Fig. R13a and g). Meanwhile, the different coupling gaps between two microwires are confirmed of 126 nm, 211 nm, 271 nm, 356 nm and 533 nm from the magnified SEM images, respectively (Fig. R13b-f). In these configurations, one microwire served as the lasing source, while the other functioned as the propagation medium.

When excited above the threshold, the leftmost microwire emits lasing, and the propagating coupled signals are detected at the terminals of the waveguide coupler to compare the PL spectra of input and output. Figure R13h shows the lasing microscope images of the waveguide couplers with different coupling gaps under the pump fluence of $62 \mu\text{J cm}^{-2}$. The observed transition from light to dark of the propagation terminals reveals the intensity of lasing propagation degenerates with the gap distance increasing. Analysis of the propagation efficiency for lasing indicates a decay trend from 38.53% to 3.23% when the gap distance increases from 126 nm to 533 nm (Fig. R13i). Therefore, a smaller coupling gap can lead to improved performance of coupled photonic devices in practical applications.

To strengthen the quality of our paper, we have added the discussion about the effect of coupling gap on coupled propagation efficiency for lasing in CsPbBr_3 waveguide couplers in Support Information, and given some explanation in the main text, as follows:

“The gap-dependent lasing waveguide-coupling measurements were also performed in two coupled microwires with different coupling gaps (Fig. S12). With the gap distance increasing, the propagation efficiency for lasing signals displays a decay trend.”

Comments 7:

It seems that the coupling efficiency between perovskite microwires is not as high as expected. In practical waveguide application, multiple stages couplings are involved. How about the coupling efficiency of multi-stage waveguides? If the coupling efficiency is low, could the authors suggest potential approaches to improve this in the future?

Response 7:

We thank the reviewer's comment. Following the reviewer's suggestion, we performed multi-stage lasing waveguide-coupling measurements between etched CsPbBr₃ microwires separated by identical coupling gaps, as shown in Fig. R14. The result displays a decay trend of the propagation efficiency for lasing *versus* the number of multi-stage waveguides.

The morphologies of CsPbBr₃ multi-stage waveguide couplers are exhibited in Fig. R14a, with the stages from 1 to 3, respectively. All the waveguide couplers are separated by identical coupling gaps of 126 nm and composed of microwires with identical dimensions of 10 $\mu\text{m} \times 2 \mu\text{m} \times 0.548 \mu\text{m}$ (Fig. R14a, c and d). In these configurations, one microwire served as the lasing source, and the others functioned as the propagation medium.

Figure R14. (a) SEM images of CsPbBr₃ multi-stage waveguide couplers with identical gaps between microwires. The microwires are of identical dimension (10 μm × 2 μm). Scale bar: 5 μm. (b) Lasing microscope images of the multi-stage waveguide couplers under the pump fluence of 62 μJ cm⁻². Scale bar: 5 μm. The white dashed circles and boxes depict excitation terminals and propagation terminals, respectively. (c) Magnified SEM image of the gap between two microwires, showing the gap of 126 nm. Scale bar: 500 nm. (d) SEM image of the side facet for one multi-stage waveguide coupler (tilted by 50° with respect to the horizontal position), indicating a height of 548 nm. Scale bar: 500 nm. Such multi-stage waveguide couplers are of identical thickness. (e) Evolution of coupled propagation efficiency for lasing as a function of the number of multi-stage waveguides at 62 μJ cm⁻², displaying a decay trend with the stage increasing.

When excited above the threshold, the leftmost microwire emits lasing, and the propagating coupled signals are detected at the terminals of the waveguide coupler to compare the PL spectra of input and output. Figure R14b shows the lasing microscope images of the multi-stage waveguide couplers under the pump fluence of 62 μJ cm⁻², and the observed transition from light to dark of the propagation terminals reveals the intensity of lasing propagation is degenerate with stage increasing. Meanwhile, analysis of the propagation efficiency for lasing indicates a decay trend from 37.8% to 0.35% corresponding to the stage from 1 to 3 (Fig. R14e).

In our work, the observed propagation loss between perovskite microwires and waveguides could be primarily attributed to the extended propagation distance, gap, and self-absorption, analogous to that in III-V-based integrated photonic circuits [*Nano Lett.* 10, 2251 (2010); *ACS Photonics* 5, 2051 (2018)]. Indeed, the lasing coupling efficiency between perovskite microwires is not as high as expected at present. From the fabrication perspective, two approaches could be suggested to improve the efficiency of the perovskite-based integrated photonic devices. On the one hand, by reducing the characteristic size of the device, optimizing the synthesis process of perovskite thin film to reduce the defect state, and adjusting the lasing mode away from the exciton state to reduce self-absorption, *etc.*, the loss of photons during the propagation process could be further reduced. On the other hand, optimizing the etching process can further improve the Q-factor of the microlaser, thereby enhancing the gain

efficiency of the integrated device. Our proposed concept can expand to the application of silicon-based optoelectronic chips in the future, *e.g.*, heterojunction integrated with silicon-based waveguide as a tunable micro light source.

To strengthen the quality of our paper, we have added the discussion about propagation efficiency for lasing in CsPbBr₃ multi-stage waveguide couplers in Support Information, and given some explanation in the main text, as follows:

“The gap-dependent lasing waveguide-coupling measurements were also performed in two coupled microwires with different coupling gaps (Fig. S12). With the gap distance increasing, the propagation efficiency for lasing signals displays a decay trend. A similar phenomenon is showcased in the multi-stage lasing waveguide-coupling measurements (Fig. S13).”.

Reply to Reviewer #3

Comments 1:

The manuscript by Qi Han et al reports the realization of integrated photonic devices based on CsPbBr₃ halide perovskites. Using focused ion beam, large area monocrystalline perovskite layers, grown by CVD on mica, are structured to obtain micrometer sized devices. Specifically, the authors demonstrate a beam splitter, Mach-Zehnder-type interferometer, and crossed-coupler. They also report characterization of lasing of a perovskite microwire and investigate the propagation efficiency of the emitted light across waveguide junctions under different coupling angles. Overall the results are a very interesting and timely as they offer proof-of-concepts for the use of CsPbBr₃ as active material in various integrated microphotonic devices. In my opinion, this is a valuable contribution and could be published after revisions. The following points should be addressed before a final decision.

Response 1:

We are grateful to the reviewer for his/her time and efforts in examining our work, as well as for the highly positive evaluations of our manuscript. We would also like to thank the reviewer for the constructive comments on how to improve our manuscript. We have made revisions following these suggestions.

Comments 2:

The features in the angle-resolved plots are difficult to distinguish with the current choice of the color scales. This is especially the case for the ARPL of the Mach-Zehnder interferometer in Fig. 5h lower panel, where one cannot really recognize much of the interference pattern from the blue background. Please correct. Also Fig. 2g showing angle-dependent lasing from the perovskite microwire could be improved.

Response 2:

We thank the reviewer for pointing out this issue, which is very helpful in improving our manuscript quality. Following the comments of the reviewer, we have adjusted the color scales of the angle-resolved photoluminescence (ARPL) spectra both in Figure 2g and Figure 5h of the main text, as shown in Fig. R15. The interference patterns of coherent lasing are more distinguishable after revision.

Figure R15. (a) Revised ARPL (left) and PL (right) spectra of the microwire above the threshold ($1.23 P_{th}$). The patterns of the ARPL spectra originate from the interference of coherent lasing modes emitted from two edges of the such microwire. **(b)** The upper panel: Normalized PL spectra of the perovskite-based MZI. The lower panel: Revised ARPL spectra of the MZI detected at C point, showing the coherence of the propagated lasing modes from two arms of the interferometer.

To strengthen the quality of our paper, we have adjusted the color scales and replaced the ARPL spectra both in Figure 2g and Figure 5h of the main text.

Comments 3:

To make the work more accessible to a broader audience of non-experts, I suggest to provide a concise, more general introduction of the basic function of the fabricated photonic devices.

Response 3:

Thank for the reviewer's suggestion. The basic functions of the fabricated photonic devices are described below:

Micro beam splitter: As a basic component in the integrated optical path, the Y-type beam splitter has a very simple structure and its main function is to realize the beam splitting and coalescence of light. An optical field (light), incident in the input port of the waveguide, is divided into two beams by the Y-branch structure and can be detected at the output end coherently. The beam splitting ratio of the Y branch is defined as $P_{\text{out1}} / P_{\text{out2}}$. The design of the Y-type beam splitter should meet the following requirements: a high propagation efficiency, a stable splitting ratio and a small device size.

Micro X-Coupler: X-Coupler is a basic component used in many kinds of photonic circuits. Cross-waveguide coupling can occur when two waveguides are brought close together and interact *via* the evanescent fields outside their boundaries. A typical X-coupler is a four-port device with two input and two output ports. In such structure, an optical field (light), incident in one of the input ports, can be split coherently into two parts through the waveguides and coupling gap and finally detected at the two output ports, respectively. Following this strategy, a transfer matrix extracted from the individual two input and two output ports can be given, as described in the equation in Figure 1b of the main text.

Mach-Zehnder interferometer (MZI): MZI is an optical device based on the principle of light interference, which is widely used to modulate the intensity and phase of output signals in fields such as optical communication, lidar, optical sensors, and so on. In the typical MZI device, the optical structure can guide the propagation direction of input light. The light signals simultaneously split along the two arms of the device respectively, and finally coalesce and interfere at the output end. By modulating the optical path difference between the two arms, the interference phase of the output signal can be tuned.

For the perovskite-based MZI in our work, one of the modulation potentials in future applications of MZI devices is the introduction of an electrical structure, *e.g.*, applying a

voltage to one arm of the device, to change the refractive index of materials and optical length, which can be used to modulate the phase and intensity of light propagation. After the coalescence of the beams from the two arms, the phase change caused by the electrical structure modulation will be transformed into the light intensity change of the output light signals, to realize the signal modulation at the output end. In future work, various hybrid optoelectronic structures can be integrated with the perovskite-based active MZI to broaden its applications.

To strengthen the quality of our paper, we have added the introduction about the basic function of the fabricated photonic devices in Support Information.

Comments 4:

On page 5 the authors write “TRPL spectrum” when instead referring to the TRPL decay dynamics. Same in the caption of Fig. 1i. Can the authors comment in more detail on the origin of the “different recombination processes” giving rise to the bi-exponential PL decay?

Response 4:

We thank the reviewer for pointing out this issue. Following the comments of the reviewer, we re-characterized and analyzed the TRPL spectra for delay dynamics of the monocrystalline CsPbBr₃ thin film, as shown in Fig. R16.

Two different lifetime components of thin film, *i.e.*, a fast component of 12.74 ns and a slow component of 96.52 ns, can be fitted by an exponential decay function, as shown in Fig. R16. Such two different time scales can be attributed to the bulk emission recombination (slow) together with the presence of surface trap states (fast), indicating that carriers can propagate deep in the material [*Science* 347, 519 (2015); *Adv. Mater.* 30, 1705992 (2018); *Adv. Opt. Mater.* 5, 1600704 (2017)]. This result agrees with the values reported in previous literature [*Adv. Mater.* 30, 1705992 (2018); *Nano Lett.* 21, 1454 (2021); *Adv. Opt. Mater.* 5, 1600704 (2017)], which suggests the high optical quality of the CsPbBr₃ thin film.

Figure R16. Time-resolved PL spectrum of the monocrystalline CsPbBr₃ thin film, the fitting decay profile (red curve) giving two lifetime components of 12.74 ns and 96.52 ns.

To strengthen the quality of our paper, we have replaced the TRPL spectrum in Figure 1i and given a more explicit description of the TRPL decay dynamics in the main text, as follows:

“TRPL delay dynamics (Fig. 1i) reveals two different lifetime components of the thin film, *i.e.*, a fast component of 12.74 ns and a slow component of 96.52 ns. Such two different time scales could be attributed to the radiative recombination of exciton (slow) together with the presence of surface trap states (fast)^{34,41,42}.”

To clarify this mechanism, we have added the corresponding references as Ref. 41 and 42.

[41] Shi, D. et al. Low trap-state density and long carrier diffusion in organolead trihalide perovskite single crystals. *Science* **347**, 519-522 (2015).

[42] Saidaminov, M. et al. Inorganic Lead Halide Perovskite Single Crystals: Phase-Selective Low-Temperature Growth, Carrier Transport Properties, and Self-Powered Photodetection. *Adv. Opt. Mater.* **5**, 1600704 (2017).

Comments 5:

Please revise the sentence at the end of page 10 (“Since perovskite semiconductors have been identified ...”). It seems something is missing.

Response 5:

We thank the reviewer for pointing out the language mistake. We have checked the sentence in the manuscript carefully and revised it following the reviewer's suggestion.

To strengthen the quality of our paper, we have revised the sentence at the end of page 10 in the main text as follows:

“Perovskite semiconductors have been identified as promising candidates for the realization of multifunctional integrated optical devices, owing to the exceptional optical quality of monocrystalline CsPbBr₃ thin films and the precision of FIB etching technology.”.

Comments 6:

There is a typo in the caption of Figure 1 on lines 3 and 4: “Mach-Zander”. Same in the Supplementary Fig. S5, last line of the caption.

Response 6:

We thank the reviewer for pointing out these language mistakes. We have checked the words in the manuscript carefully and revised them following the reviewer's suggestions.

To strengthen the quality of our paper, we have revised the caption words both in Figure 1 and Supplementary Materials as “Mach-Zehnder” in the revised version.

Comments 7:

The acronyms (FIB, PL, FP, MZI, SEM, ...) are defined multiple times throughout the manuscript.

Response 7:

We thank the reviewer for pointing out these acronym problems. We have checked all the acronyms in the manuscript carefully and revised the redefined ones following the reviewer's suggestion.

To strengthen the quality of our paper, we have revised all the redefined acronyms in the revised version.

Comments 8:

Finally, in the Discussion section, I recommend discussing in more detail the relevance of the demonstrated perovskite photonic devices also in the framework of efforts towards nanophotonic design strategies to engineer light-matter interaction.

Response 8:

We thank the reviewer's suggestion. It is indeed vital to the development of perovskite photonic devices that the design and fabrication of sub-wavelength photonic structures. We think that nanophotonics could be developed in the following fields:

- (1) Regarding micro/nano laser, employing the structures of bound-states-in-the-continuum (BIC) photonic crystals, vortex laser and ultrahigh-Q laser can be realized to act as high-coherent light sources in photonic chips and optical communications [*Science* 377, 1215 (2022), *Science* 367, 1018 (2020); *Nat. Photonics* 14, 623, (2020)]. Moreover, the micro/nano laser with high locality and high coherence can also be generated by moiré photonic crystals [*Science* 362, 1153 (2018); *Nat. Nanotechnol.* 16, 1099 (2021)].
- (2) Regarding propagation manipulation, various metasurfaces and Bloch surface wave structures could demonstrate the highly efficient in-plane propagation and waveguide coupling of light [*Nat. Nanotechnol.* 13, 906 (2018); *Laser Photonics Rev.* 17, 2201001 (2023)]. In particular, polarized light emission could be manipulated to couple with different channel waveguides directionally and selectively *via* designed metasurfaces [*Nano Lett.* 23, 8, 3326 (2023)].

(3) Regarding light-matter interaction, utilizing the strong coupling between excitons and BIC microcavities, the ultra-low threshold polariton lasing and Bose-Einstein condensation can be achieved in microcavities with BIC structures, which is useful for studying macroscopic quantum phenomena in room-temperature solid systems [*Nature* **605**, 447 (2022); *Nat. Mater.* **22**, 964 (2023)].

While there are great challenges in sub-wavelength nanofabrication processes and integration, the potential for nanophotonics research and applications is promising and huge.

To strengthen the quality of our paper, we have added the following sentences in the main text:

“In the future, integrated photonics will develop towards subwavelength nanostructures. Bound-states-in-the-continuum and moiré photonic crystals could facilitate on-chip vortex and high-Q laser sources^{48,49}. Light-matter strong coupling could also generate room-temperature ultralow threshold polaritonic lasers in nanocavities⁵⁰. Metasurface structures could manipulate highly efficient in-plane propagation and coupling of light⁵¹. The potential of nano and micro photonic devices is promising for advancing information and computing science.”.

And we have also added the corresponding references as Ref. 48-51.

[48] Huang, C. et al. Ultrafast control of vortex microlasers. *Science* **367**, 1018-1021 (2020).

[49] Mao, X. et al. Magic-angle lasers in nanostructured moiré superlattice. *Nat. Nanotechnol.* **16**, 1099-1105 (2021).

[50] Ardizzone, V. et al. Polariton Bose-Einstein condensate from a bound state in the continuum. *Nature* **605**, 447-452 (2022).

[51] Chen, Y. et al. Efficient meta-couplers squeezing propagating light into on-chip subwavelength devices in a controllable way. *Nano Lett.* **23**, 3326-3333 (2023).

REVIEWER COMMENTS

Reviewer #1 (Remarks to the Author):

The authors have addressed most of my concern, and the manuscript has been improved quite a lot. I still have some additional comments, which may help the authors to further strengthen the manuscript.

1. The discussion of TRPL decay is not sufficient and lack of direct evidence. One cannot simply attribute the two carrier lifetimes to bulk and surface recombinations. If it is true, the authors need to directly measure the trap concentration in the bulk and on the surface. In my opinion, high bimolecular recombination is more likely to be the reason of the fast decay in the beginning. So, I suggest the authors to perform light intensity dependent TRPL and provide the excitation fluence (or generation rate) of TRPL measurements.

2. I do not agree that carrier lifetime could be ignored for averaged pulses. Firstly, the pulse laser pumped devices are not operating as steady state devices, and the multiple pulses averaged signal only improve the signal-to-noise ratio (SNR). Secondly, even for steady-state measurements, carrier lifetime is one of the most important parameters, which affect the optical gain [Semiconductor Optoelectronics (Farhan Rana, Cornell University), chapter 11 Basics of Semiconductor Lasers].

3. There are small grains on the surface of the perovskite patterns, which can be observed from the SEM images. It seems that perovskite cavities are not composed of perovskite single crystal. The authors need to provide more evidence if they want to claim the devices were fabricated based on single crystals, such as single crystal XRD.

Reviewer #2 (Remarks to the Author):

The authors answered all my questions and have made corrections of the manuscript. I have no questions any more, I recommend the acceptance of this paper.

Reviewer #3 (Remarks to the Author):

The authors have provided additional information in response to the referees and implemented corresponding modifications in the manuscript and in particular in the revised supplementary information. I have no further comments and can recommend the revised manuscript for publication.

Response to Reviewers' Comments and Revised Details

Manuscript ID: NCOMMS-23-15734A

Reply to Reviewer #1

Comments 1:

The authors have addressed most of my concern, and the manuscript has been improved quite a lot. I still have some additional comments, which may help the authors to further strengthen the manuscript.

Response 1:

We are grateful to reviewer #1 for his/her time and efforts in examining our work, as well as for the positive comments on our manuscript. Reviewer #1 provided further constructive suggestions for improving our paper, and we have carefully accommodated his/her comments and performed corresponding experiments in the revised manuscript, which will be discussed below.

Comments 2:

The discussion of TRPL decay is not sufficient and lack of direct evidence. One cannot simply attribute the two carrier lifetimes to bulk and surface recombinations. If it is true, the authors need to directly measure the trap concentration in the bulk and on the surface. In my opinion, high bimolecular recombination is more likely to be the reason of the fast decay in the beginning. So, I suggest the authors to perform light intensity dependent TRPL and provide the excitation fluence (or generation rate) of TRPL measurements.

Response 2:

We thank the reviewer for his/her comment and suggestion. Following the suggestion, we have performed the measurement of pump-fluence dependent time-resolved photoluminescence (TRPL) for CsPbBr₃ thin films. We analyzed the experimental TRPL data and concluded the fast lifetime and long lifetime originating from the bimolecular and intrinsic exciton recombination process, respectively, which agrees with the reviewer's opinion.

TRPL spectra of CsPbBr₃ thin films have been measured under pump fluences ranging from 2.8 $\mu\text{J cm}^{-2}$ to 27.5 $\mu\text{J cm}^{-2}$, as shown in Fig. R1a. The results of Fig. R1a are fitted by two exponential decay functions, then a shorter lifetime (τ_1) and a longer lifetime (τ_2) can be obtained. Figures R1b and R1c show the evolution of the two extracted lifetimes (τ_1 and τ_2) and two corresponding weights (A_1 and A_2) as a function of pump fluence, respectively. To compare the change of lifetime, we set the lifetime components under the initial pump fluence of 2.8 $\mu\text{J cm}^{-2}$ as τ_{01} and τ_{02} , respectively, and calculated the corresponding lifetime ratio. One can observe that both τ_1 and τ_2 decrease with pump fluence increasing. Notably, with pump fluence rising, **the weight of τ_1 increases** and gradually becomes dominant in the TRPL spectrum, on the contrary, **the weight of τ_2 reduces**. **The weight ratio of A_1/A_2 gradually increases and eventually tends to saturation** (Fig. R1d).

These results indicate that the longer lifetime τ_2 originates from the radiative recombination of intrinsic exciton, and τ_1 comes from the bimolecular recombination process of excitons. As pump fluence rises, the increasing carrier density leads to the formation of a higher-order two-body recombination process and a stronger interaction. This two-body process is positively correlated to the carrier density and is dominant at high pump fluence. Thus, the proportion of bimolecular recombination in TRPL shows an increasing trend at higher excitation density. The intrinsic exciton recombination process tends to saturation when the pump power exceeds the critical point. Our results

of dynamics measurement agree with previous literature [*Nat. Photonics* 8, 737 (2014); *Nat. Commun.* 8, 14558 (2017)].

Figure R1. (a) Time-resolved PL spectra for the monocrystalline CsPbBr₃ thin film under various pump fluences from 2.8 μJ cm⁻² to 27.5 μJ cm⁻². (b) The evolution of the two different extracted lifetime components τ_1 and τ_2 as functions of pump fluence, setting the lifetime components under the initial pump fluence of 2.8 μJ cm⁻² as τ_{01} and τ_{02} , respectively. (c, d) The evolution of the weights (c, A₁ and A₂) and weight ratio (d, A₁/A₂) of the two different lifetime components τ_1 and τ_2 as functions of pump fluence, respectively.

To strengthen the quality of our paper, we have revised the description of the TRPL decay dynamics in the main text, as follows:

“Such two different time scales could be attributed to the radiative recombination of intrinsic excitons (slow) and the bimolecular recombination process of excitons (fast) in the system^{42,43}.”

We have cited the references of Ref. 41 and 42 to explain the corresponding mechanism.

[42] S. Manser, J., V. Kamat, P. Band filling with free charge carriers in organometal halide perovskites. *Nat. Photonics* **8**, 737-743 (2014).

[43] Xing, G. et al. Transcending the slow bimolecular recombination in lead-halide perovskites for electroluminescence. *Nat. Commun.* **8**, 14558 (2017).

Comments 3:

I do not agree that carrier lifetime could be ignored for averaged pulses. Firstly, the pulse laser pumped devices are not operating as steady state devices, and the multiple pulses averaged signal only improve the signal-to-noise ratio (SNR). Secondly, even for steady-state measurements, carrier lifetime is one of the most important parameters, which affect the optical gain [Semiconductor Optoelectronics (Farhan Rana, Cornell University), chapter 11 Basics of Semiconductor Lasers].

Response 3:

We thank the reviewer's comment and agree with his/her opinion that carrier lifetime is one of the most important parameters in an optical lasing, which would affect the optical gain and transient electric field distribution.

The perovskite microwire laser demonstrated in our work is excited by a femtosecond pulsed laser (wavelength: 400 nm, repetition rate: 1kHz, pulse width: 100 fs). Therefore, two parameters of time need to be considered when investigating the detailed transient lasing property in the microstructure, *i.e.*, the carrier intrinsic lifetime of the material and the pulse duration time of the pumping laser.

To explain such a mechanism, firstly, a streak camera was used to characterize the carrier lifetime of the perovskite microwire above the threshold ($1.2 P_{th}$), as shown in Fig. R2. Notably, the stimulated emission with an ultrashort decay time dominates the emission process above the lasing threshold, compared to the spontaneous emission of carriers. The extracted corresponding lasing lifetime is 8 ps, much longer than the duration time (100 fs) of a single pulse for the femtosecond laser. During the excitation

of a single pulse and carrier lifetime, there is enough time and density to keep efficient optical gain to form population inversion in the perovskite microwire, so that the system can support the formation of a steady standing wave distribution and lasing oscillation modes. In such a scenario, the influence of carrier lifetime on the transient lasing property can be reconciled due to the matching time window between the lasing lifetime of the system and the pulse duration time of the pumping laser, in which sufficient gain carriers can maintain the steady state of population inversion.

Thus, the perovskite microwire laser we demonstrated here can be considered as a 1kHz integrated microlaser source with picosecond pulse width. In the future, ultrafast all-optical modulations with picosecond scale are promising to be applied on the perovskite-based active multifunctional integrated photonic devices in our work.

Figure R2. (a) Streak camera image of an etched CsPbBr₃ microwire above the threshold at 1.2 P_{th}. (b) Extracted TRPL spectrum (blue circles) of the lasing emission from yellow dash line of (a), fitted by an exponential decay function (red line), giving an ultrashort lifetime of about 8.0 ps.

Meanwhile, the lasing carrier density of the perovskite microwire above the threshold (1.2 P_{th}) can be estimated as follows:

Average pump fluence: 90.7 nW, the energy of a single pulsed excitation lasing ($\lambda = 400$ nm, 1 kHz, 100 fs): $90.7 \times 10^{-9} \times 1/1000$ J = 9.07×10^{-11} J,

The number of photons contained in a single pulse:

$$\frac{9.07 \times 10^{-11}}{\frac{1240}{400} \times 1.6 \times 10^{-19}} = 1.83 \times 10^8,$$

The duration of a 400 nm single pulsed excitation lasing is 100 fs, and the decay time of stimulated emission above the threshold is about 8.0 ps, so the effective number of photons in the lifetime of carrier is 1.83×10^8 .

Considering the gain volume of CsPbBr₃ microwire ($10 \mu\text{m} \times 2 \mu\text{m} \times 0.4 \mu\text{m}$) and photoluminescence quantum yield, we can thus give a reasonable estimate of the gain carrier density above the threshold ($1.2 P_{\text{th}}$): $2.34 \times 10^{18} \text{ cm}^{-3}$.

Such sufficient gain carrier density can effectively support the generation of photonic lasing [*Nat. Commun.* 10, 265 (2019); *ACS Photonics* 7, 454 (2020)].

Comments 4:

There are small grains on the surface of the perovskite patterns, which can be observed from the SEM images. It seems that perovskite cavities are not composed of perovskite single crystal. The authors need to provide more evidence if they want to claim the devices were fabricated based on single crystals, such as single crystal XRD.

Response 4:

Thank the reviewer for pointing out this issue. To further prove the monocrystalline property of the sample, we have performed micro-region X-ray diffraction (XRD) measurements on the CsPbBr₃ thin films. Meanwhile, the more precise XRD measurements with high resolution help us to correct the analysis of crystal structure in the main text and strengthen the quality of our manuscript.

The observed small grains on the surface of perovskite patterns come from the sprayed inhomogeneous gold particles during the scanning electron microscopy (SEM) measurement. Perovskite thin films grow on the mica substrate, which is insulating. In order to reduce the charging effect during the SEM testing, the sample surface needs to

be sprayed with a thick gold layer of several nanometers and thus would introduce additional gold particles in the microstructure area.

In our previous macro XRD measurement of the CsPbBr₃ thin films, the results could not distinguish the tiny splitting of diffraction peaks, due to the low resolution of the test step. Thus, the previous XRD analysis in the main text were labeled as cubic phase, which are incorrect. Here, we carried out more precise XRD measurements both on large-area (millimeter size) and micro-region (a range of 200 micrometers), respectively, to further prove the monocrystalline property and identify the crystal structure, as shown in Fig. R3a and R3b. One can observe that tiny splitting XRD peaks explicitly appear at ~15.2° and 30.7° both in Fig. R3a (macroscale) and R3b (microscale), corresponding to the orthorhombic phase structure of CsPbBr₃ (ICSD #97851). This result reveals the good monocrystalline properties of the thin films. It should be noted that the splitting at the (002)/(110) and (004)/(220) peaks are direct evidence to exclude the cubic phase of CsPbBr₃ [ACS Nano 13, 10085 (2019); ACS Nano 14, 15605 (2020)].

Figure R3. (a) Large-area (millimeter size) and (b) Micro-region (a range of 200 micrometers) XRD of a monocrystalline CsPbBr₃ thin film, showing the orthorhombic phase structure and good monocrystalline properties. The XRD peaks originating from the pure mica substrate are marked by *. The insets show the magnified tiny splitting XRD peaks of the sample at ~ 30.7°.

To strengthen the quality of our paper, we have replaced the XRD analysis in Figure 1g and added the micro-region XRD of the monocrystalline CsPbBr₃ thin film in

Supplementary materials. The corresponding discussion in the main text has been revised as:

“As shown in the lower panel of Fig. 1g, tiny splitting XRD diffraction peaks of the sample at ~ 15.2° and 30.7° are indexed to the orthorhombic phase structure (ICSD #97851), revealing the excellent crystal quality of the thin film without any impurity peaks from CsBr or PbI₂ (Fig. S2)^{30,41}. Moreover, a high-resolution TEM image of the cross-section for CsPbBr₃ thin film demonstrates the lattice spacing of 0.29 nm, also showing the brilliant monocrystalline property (Fig. S2).”

We have also added the corresponding reference as Ref. 52.

[41] Zhao, L. et al. Vapor-phase incommensurate heteroepitaxy of oriented single-crystal CsPbBr₃ on GaN: toward integrated optoelectronic applications. *ACS Nano* **13**, 10085-10094 (2019).

Reply to Reviewer #2

Comments 1:

*The authors answered all my questions and have made corrections of the manuscript.
I have no questions any more, I recommend the acceptance of this paper.*

Response 1:

We thank the reviewer for the positive recommendation.

Reply to Reviewer #3

Comments 1:

The authors have provided additional information in response to the referees and implemented corresponding modifications in the manuscript and in particular in the revised supplementary information. I have no further comments and can recommend the revised manuscript for publication.

Response 1:

We are grateful to the reviewer for the positive recommendation.

REVIEWERS' COMMENTS

Reviewer #1 (Remarks to the Author):

I have no further comments and suggestions.